# Histone chaperone ASF1 mediates H3.3-H4 deposition in Arabidopsis

Zhenhui Zhong [1,10], Yafei Wang[2,10], Ming Wang[1,10], Fan Yang[2], Quentin Angelo Thomas [3], Yan Xue [1], Yaxin Zhang [4], Wanlu Liu [5], Yasaman Jami-Alahmadi[6], Linhao Xu [7], Suhua Feng[1,8], Sebastian Marquardt [3], James A. Wohlschlegel[6], Israel Ausin [2] ✉ & Steven E. Jacobsen [1,9] ✉

Histone chaperones and chromatin remodelers control nucleosome dynamics, which are essential for transcription, replication, and DNA repair. The histone chaperone Anti-Silencing Factor 1 (ASF1) plays a central role in facilitating CAF-1-mediated replication-dependent H3.1 deposition and HIRA-mediated replication-independent H3.3 deposition in yeast and metazoans. Whether ASF1 function is evolutionarily conserved in plants is unknown. Here, we show that Arabidopsis ASF1 proteins display a preference for the HIRA complex. Simultaneous mutation of both Arabidopsis *ASF1* genes caused a decrease in chromatin density and ectopic H3.1 occupancy at loci typically enriched with H3.3. Genetic, transcriptomic, and proteomic data indicate that ASF1 proteins strongly prefers the HIRA complex over CAF-1. *asf1* mutants also displayed an increase in spurious Pol II transcriptional initiation and showed defects in the maintenance of gene body CG DNA methylation and in the distribution of histone modifications. Furthermore, ectopic targeting of ASF1 caused excessive histone deposition, less accessible chromatin, and gene silencing. These findings reveal the importance of ASF1-mediated histone deposition for proper epigenetic regulation of the genome.

In eukaryotes, DNA is compacted and organized into chromatin. The basic unit of the chromatin is the nucleosome, composed of 147 base pairs of DNA wrapped around a core histone H3-H4 tetramer and two histone H2A-H2B dimers. The organization of DNA into chromatin is essential for genomic compaction and gene expression regulation[1,2]. However, it also presents a barrier to transcription[3]. Therefore, proper transcription requires dynamic nucleosome compositions and

positioning in a process involving frequent deposition and eviction of histones spatially and temporally. This highly dynamic process is assisted by histone chaperones, a group of histone binding proteins that play crucial roles in nucleosome formation.

Histone H3 has several variants in mammals and plants. For instance, mammals contain the canonical H3.1 and H3.2 variants (that only differ in a Cys96-to-Ser96 substitution) and the replacement

[1]Department of Molecular, Cell and Developmental Biology, University of California, Los Angeles, CA 90095, USA. [2]State Key Laboratory of Crop Stress Biology for Arid Areas, College of Life Sciences and Institute of Future Agriculture, Northwest A&F University, Yangling 712100 Shaanxi, China. [3]Copenhagen Plant Science Centre, Department of Plant and Environmental Sciences, University of Copenhagen, Frederiksberg, Denmark. [4]Haixia Institute of Science and Technology, Fujian Agriculture and Forestry University, Fuzhou 350002 Fujian, China. [5]Zhejiang University-University of Edinburgh Institute (ZJU-UoE Institute), Zhejiang University School of Medicine, International Campus, Zhejiang University, 718 East Haizhou Road, Haining 314400 Zhejiang, China. [6]Department of Biological Chemistry, University of California, Los Angeles, CA 90095, USA. [7]Leibniz Institute of Plant Genetics and Crop Plant Research (IPK), OT Gatersleben, Stadt Seeland 06466, Germany. [8]Eli & Edythe Broad Center of Regenerative Medicine & Stem Cell Research, University of California, Los Angeles, CA 90095, USA. [9]Howard Hughes Medical Institute, University of California, Los Angeles, CA 90095, USA. [10]These authors contributed equally: Zhenhui Zhong, Yafei Wang, Ming Wang. ✉e-mail: israel.ausin@gmail.com; jacobsen@ucla.edu

variant H3.3. Although there is a difference of only 4-5 amino acids between H3.1/2 and H3.3, these two variants generally exhibit opposing genomic distributions, different functions, and distinct post-translational modifications[4–6]. Similarly, in Arabidopsis, H3.1 and H3.3 only differ in 4 amino acids[7] and also show distinct genomic patterns and bear different post-translational modifications. For instance, H3.1 is enriched in heterochromatin and is associated with repressive chromatin marks[8,9]. Like in mammals, its deposition occurs during S-phase in a replication-dependent manner at replication forks of newly synthesized DNA and is controlled by the Chromatin Assembly Factor-1 (CAF-1) complex[10,11]. In Arabidopsis, the CAF-1 complex is comprised of FASCIATA1 (FAS1), FASCIATA2 (FAS2), which are specific to the CAF-1 complex, and MULTICOPY SUPPRESSOR OF IRA1 (AtMSI1)[12,13]. Although *fas* mutants exhibit several morphological and molecular defects[14–16], they produce viable progeny[17], and incorporation of H3.1 is only partially abolished[18,19]. Indeed, most of the *fas2-4* syndrome is rescued by downregulating the Salicylic acid (SA) pathway, indicating that the pleiotropism might be mostly an effect of constitutive activation of the SA pathway rather than a massive loss of H3.1 deposition[17].

On the other hand, Arabidopsis H3.3 is often found in transcribed regions and is associated with chromatin features typical of gene activation[8,9,20–22]. H3.3 deposition is mainly performed by the histone regulator A (HIRA) complex and occurs throughout the cell cycle in a replication-independent manner[10]. Like in mammals, the Arabidopsis HIRA complex includes four subunits, CABIN, HIRA, UBINUCLEIN1 (UBN1), and UBN2[23,24]. Like *fas* mutants, the *hira-1* mutant is fertile and only shows minor developmental defects, including short hypocotyls and roots. Moreover, the quadruple mutant *cabin1-1 hira-1 ubn1-1 ubn2-1* phenotype is similar to the single mutant *hira-1*[24]. The fact that *fas1-4 hira-1* double mutants display a more severe phenotype than the single mutants suggests some degree of redundancy between the CAF-1 and the HIRA pathways[23].

In mammals, the heterodimer formed by Alpha Thalassemia-mental Retardation X-linked (ATRX) and death domain-associated protein (DAXX) is also involved in the deposition of H3.3 independently of HIRA[25]. Although DAXX homologs have not been found in plant genomes, the Arabidopsis ATRX homolog appears to be involved in a HIRA-independent H3.3 deposition pathway, given that mutation of *ATRX* causes a reduction in H3.3 occupancy[26,27].

Elevated levels of newly synthesized histones in the cytoplasm are toxic[28] and must be transported to the nucleus to be incorporated into chromatin. In mammals, this function is carried out by Nuclear Auto-antigenic Sperm Protein (NASP) and histone chaperone Anti-Silencing Factor 1 (ASF1) proteins by binding H3-H4 dimers and transferring them to HIRA and CAF-1[29–31]. Arabidopsis NASP can also bind H3-H4 dimers and H3 monomers in vitro[32] and H3.1 or H3.3 in vivo[33]. Interestingly, Arabidopsis NASP shifts the H3-H4 dimer-tetramer equilibrium towards the tetramer in vitro, suggesting that it could act in part by creating preassembled H3-H4 tetramers[32].

In yeast and metazoans, ASF1 plays a central role in the replication-dependent H3.1 and replication-independent H3.3 deposition by transferring H3.1-H4 and H3.3-H4 dimers to CAF-1 or HIRA, respectively[29,31,34,35]. Most vertebrates carry two ASF1 homologs; in mammals, ASF1 paralogs (termed ASF1a and ASF1b) have different functions and exhibit distinct expression patterns[36,37]. ASF1a is constitutively expressed throughout the cell cycle and forms a heterodimer with the B-domain of HIRA in the nucleus, whereas ASF1b is specifically expressed in S-phase and interacts with the CAF-1 complex[37,38]. In Arabidopsis, ASF1 has two functionally redundant homologs, ASF1A and ASF1B, expressed throughout the cell cycle. Depletion of either ASF1A or ASF1B does not cause apparent morphological defects, whereas the simultaneous mutation of *ASF1A* and *1B* significantly inhibits plant growth, affects reproductive organ development, alters the response to heat stress, and affects chromatin

assembly during replication[39,40]. However, it is not known whether plant ASF1 proteins play similar functions as their homologs in mammals. Here, we show that the *asf1a-1 asf1b-1* double mutant displays lower nucleosome occupancy than the wild type, *fas1-4*, and *fas2-4* at loci typically enriched with H3.3 over H3.1. Furthermore, ASF1 proteins localize at these loci. We show that ASF1 proteins strongly interact with the HIRA complex in vivo, as assayed by immunopurification followed by Mass Spectrometry and Co-Immunoprecipitation. Surprisingly, we did not find any evidence of interaction with any component of the Arabidopsis CAF-1 complex. Consistently, genetic analysis showed an epistatic relationship between *ASF1A*, *1B*, and *HIRA*, while the triple mutant *asf1a-1 asf1b-1 fas2-4* appears to be lethal. Finally, *asf1a-1asf1b-1* mutants show epigenetic features that are consistent with a role in facilitating HIRA-mediated histone deposition, including cryptic Pol II initiation and defects in the distribution of histone post-translation modifications and CG gene body DNA methylation. Ectopic targeting of ASF1 also caused gene silencing associated with increased histone occupancy and decreased chromatin accessibility. Together these data suggest that in Arabidopsis, ASF1s are essential in supplying histone variants to the HIRA complex, which plays critical roles in epigenetic gene regulation.

## Results

### The *asf1a1b* double mutant alters chromatin accessibility in a pattern resembling that of the *hira-1* mutant

The CAF-1 and HIRA deposition pathways are essential for depositing their corresponding histone H3 variants at specific regions of the genome, which would explain the developmental abnormalities of *fas* and *hira-1* mutants. *asf1a-1 asf1b-1* (hereafter *asf1a1b*) mutants also show developmental abnormalities[40], suggesting that ASF1 activity is also likely important for histone deposition at specific regions. To find these specific regions, we started by profiling chromatin accessibility in wild type (Col-0), *asf1a1b*, *fas1-4*, *fas2-4*, and *hira-1* mutants by Assay for Transposase-Accessible Chromatin (ATAC-Seq) (Supplementary Fig. 1a, b)[41]. First, we examined whole-genome variation in chromatin accessibility. As expected, given the known distribution of H3.1 and H3.3, we found that compared to wild type, *hira-1* mutants showed higher chromatin accessibility in euchromatic regions, while *fas* mutants showed higher accessibility in pericentromeric heterochromatin (Supplementary Fig. 1c–e). The highly accessible regions detected in *hira-1* and *fas* mutants generally correlated with the distribution of H3.3 and H3.1, respectively (Supplementary Fig. 1f, g). Strikingly, the *asf1a1b* pattern resembled that of *hira-1* but not that of *fas1-4* or *fas2-4* (Supplementary Fig. 1h). We next examined chromatin accessibility over transposable elements (TEs) and protein-coding genes. Consistent with previous data, *fas* mutants showed slightly higher chromatin accessibility over TEs (mainly located near the centromeres) than the wild type, whereas *asf1a1b* and *hira-1* exhibited similar levels compared to wild type (Supplementary Fig. 2a–e). General patterns of chromatin accessibility over gene bodies were roughly similar for all genotypes tested. However, metagene analyses revealed higher chromatin accessibility at the 3′-end in *asf1a1b* and *hira-1* mutants and lower chromatin accessibility towards the 5′-end that was specific for *asf1a1b* (Fig. 1a), while we did not detect substantial changes in *fas* mutants compared to wild type (Fig. 1a, and Supplementary Fig. 2f, g). *asf1a1b*-specific highly accessible regions overlapped with those of *hira-1* and corresponded to H3.3-enriched and H3.1-depleted regions (Fig. 1b–d and Supplementary Fig. 2h).

### ASF1s genetically and physically interact with HIRA

To determine the interaction partners of ASF1 proteins in Arabidopsis, we performed immunoprecipitation in combination with mass spectrometry (IP-MS) using *pASF1A::ASF1A-3xFlag-BLRP* and *pASF1B::ASF1B-9xMyc-BLRP* complementing lines. In both cases, we identified AtNASP, HIRA, and all the known members of the HIRA complex, including

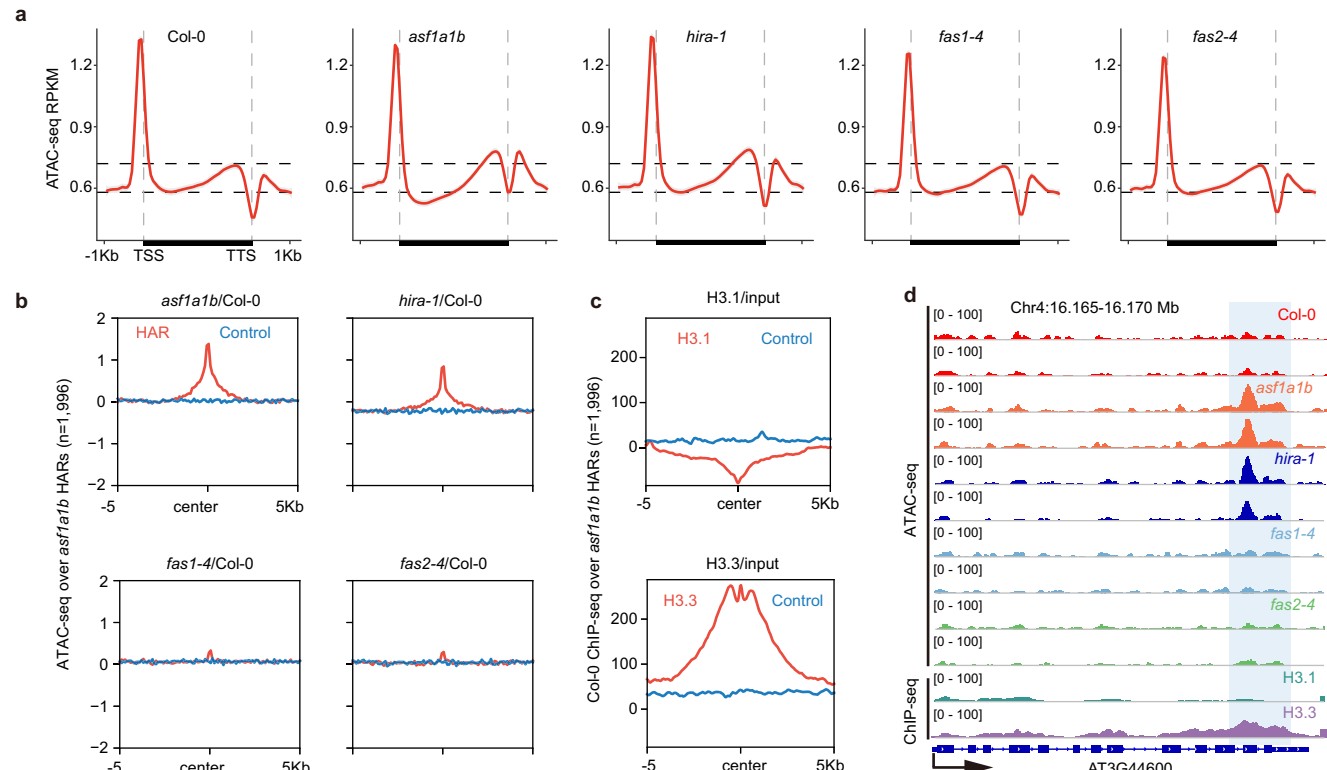

**Fig. 1 | Effect of H3 Histone chaperone mutants on chromatin accessibility.**
**a** Metaplot of ATAC-seq signals of Col-0, *asf1a1b*, *hira-1*, *fas1-4*, and *fas2-4* over the gene body, including 1 Kb flanking sequences of genes with coding regions lengths over 150 bp (*n* = 26816). **b** Metaplot of ATAC-seq signals of *asf1a1b*/Col-0, *hira-1*/Col-0, *fas1-4*/Col-0, *fas2-4*/Col-0 over *asf1a1b* highly accessible regions (HAR, *n* = 1996). The blue line represents metaplot of randomly selected control regions of *asf1a1b* highly accessible regions. **c** Metaplot of Col-0 ChIP-seq signals of H3.1/input, H3.3/input over *asf1a1b* highly accessible regions (HAR, *n* = 1996). The blue line represents a metaplot of randomly selected control regions of *asf1a1b* highly accessible regions. Two experiments were repeated with similar results. **d** Representative example showing chromatin accessibility of *asf1a1a*, *hira-1*, *fas1-4*, and *fas2-4* over the higher accessible peak.

CABIN1, UBN1, and UBN2 (Supplementary Table 1, Fig. 2a, b). Furthermore, we confirmed the ASF1s-HIRA interaction by co-immunoprecipitation (co-IP) using a complementing *pHIRA::HIRA-3xHA* line (Fig. 2c). We also detected TOUSLED, a conserved serine/threonine nuclear kinase that phosphorylates ASF1a and histone H3 in humans[42] in both IP-MS datasets (Supplementary Table 1, Fig. 2a, b). Apart from the kinase activity, Tousled-Like kinase 2 also acts as an ASF1 regulator by mimicking H3 and recognizing ASF1's H3-H4 binding pocket[43]. Although the two ASF1 proteins shared most of the interacting partners, we did not observe physical interaction between them, suggesting that in Arabidopsis, two mutually exclusive HIRA complexes may exist depending on whether they include ASF1A or ASF1B. Furthermore, consistent with our ATAC-Seq data analysis, we did not detect any peptides from components of the CAF-1 complex (FAS1, FAS2, or AtMSI1). To validate the hypothesis that ASF1 works with HIRA but not with the CAF1 complex, we crossed *asf1a1b* with *fas* or *hira-1* mutants. *asf1a1b hira-1* triple mutants were nearly identical to *asf1a1b* double mutants, indicating an epistatic relationship between *HIRA* and *ASF1s*, whereas we could not recover an *asf1a1b fas1-4* nor *asf1a1b fas2-4* triple mutant indicating that this genetic combination is likely to be lethal (Fig. 2d, e, Supplementary Fig. 3 and Supplementary Table 2). Together, these results show that ASF1 proteins form a complex with HIRA and that the CAF-1 complex is likely independent of ASF1s.

## ASF1 proteins physically interact with H3.1 and H3.3 in vivo
IP-MS experiments did not yield a significant number of peptides from H3.1 and H3.3. To determine the interaction of ASF1 proteins with H3 variants, we performed co-IP assays using *ASF1A* and *1B-9xMyc* complementing lines, and H3.1 and H3.3 3xFlag tagged lines. As shown in Supplementary Fig. 4a, we detected an interaction of ASF1 proteins

with both histone variants. Since AtNASP peptides were present in our IP-MS experiments, these data are consistent with a previous study showing that AtNASP can interact with H3.1 and H3.3[32].

## ASF1 mutation causes a redistribution of H3.3 and H3.1
To corroborate that the alterations in chromatin accessibility observed in *asf1a1b* were due to defective H3 deposition, we conducted histone H3.1 and H3.3 chromatin immunoprecipitation followed by sequencing (ChIP-seq) in the wild type and *asf1a1b* mutants. Metagene analysis indicated that compared to wild type, *asf1a1b* showed decreased H3.3 and increased H3.1 occupancy at the 3' end of genes (Supplementary Fig. 4b). However, we observed the opposite trend in TEs, a gain of H3.3 and a loss of H3.1 (Supplementary Fig. 4c). This constitutes a genome-wide redistribution of H3.1 and H3.3, suggesting that ASF1 proteins are essential for the proper distributions of these two H3 variants (Fig. 3a, b, and Supplementary Fig. 4d).

To further analyze the alteration of H3 variants in *asf1a1b*, we plotted them over H3.1-unique, H3.3-unique, and H3.1/H3.3-common peaks (according to their distribution in the wild type). At the H3.3-unique peaks, we observed a decrease in the H3.3 signal and a corresponding increase in the H3.1 signal, consistent with the role of ASF1 in depositing H3.3 via the HIRA complex (Fig. 3c). Interestingly, although H3.1 appears to at least partially replace the missing H3.3 in these regions, the overall chromatin accessibility was increased (Supplementary Fig. 4e). At H3.1 unique peaks, we observed the opposite pattern, an increase in H3.3 and a decrease in H3.1 in *asf1a1b* (Fig. 3d). In addition, these regions showed lower chromatin accessibility (Supplementary Fig. 4e). At the H3.1/H3.3-common peaks, we observed only a moderate increase of H3.1 and a slight decrease of H3.3 (Fig. 3e). Together these data show that ASF1 mutation causes a

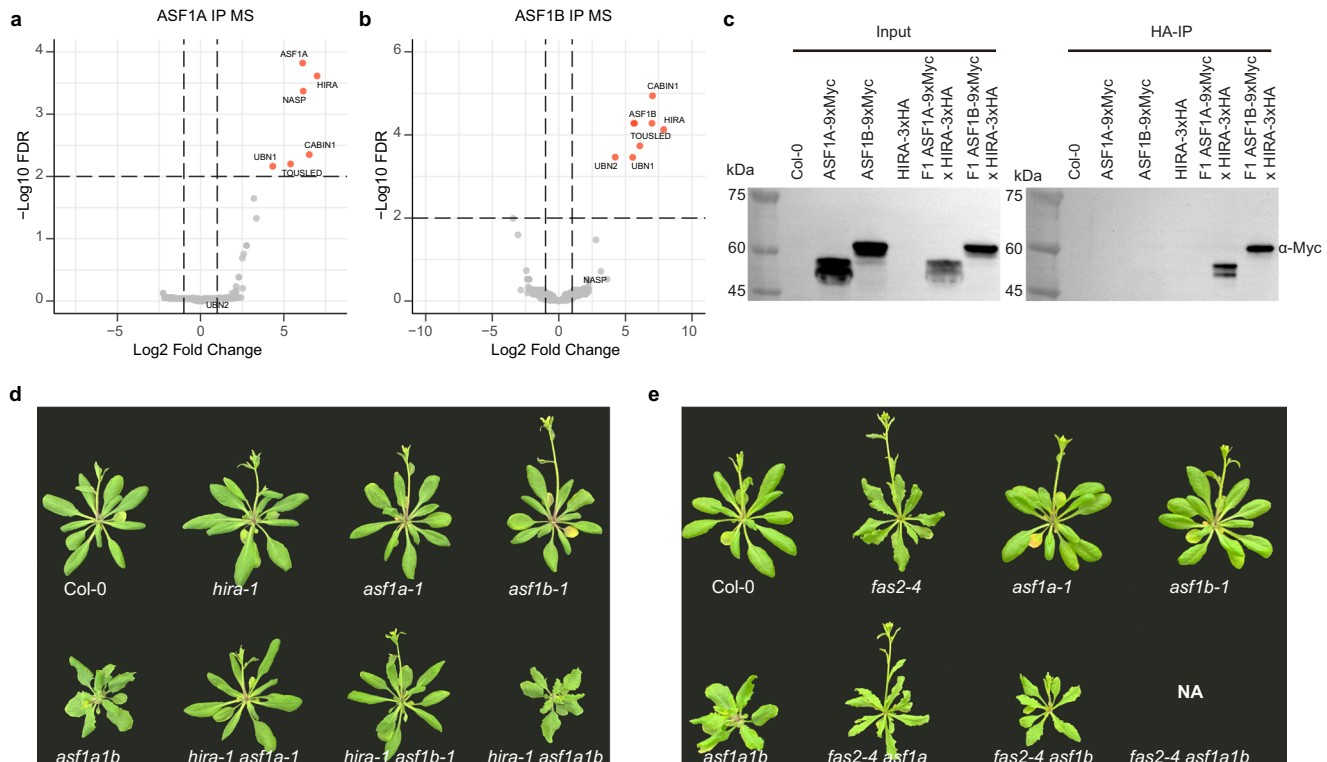

**Fig. 2 | Genetic relationships between H3 Histone chaperone mutants.** Volcano plots showing proteins interact with ASF1A (**a**) and ASF1B (**b**) identified by IP-MS. The distribution of identified proteins according to the Log2 fold change of average spectral counts and −Log10 FDR values. Proteins with FDR < 0.01 are in red. **c** Co-immunoprecipitation assay of ASF1A, ASF1B, and HIRA. Labels to the right indicate the antibody used for the western blot. **d** Morphology of Col-0, *hira-1, asf1a-1, asf1b-1, asf1a1b, hira-1 asf1a-1, hira-1 asf1b-1*, and *hira-1 asf1a1b*. **e** Morphology of Col-0, *fas2-4, asf1a-1, asf1b-1, asf1a1b, fas2-4 asf1a-1*, and *fas2-4 asf1b-1*. NA: *fas2-4 asf1a1b* line could not be obtained.

redistribution of H3.3 and H3.1 in the genome and corresponding changes in chromatin accessibility, such that higher levels of H3.3 correspond to lower levels of chromatin accessibility. Given that the IP-MS and genetic data indicate that ASF1 works with HIRA rather than with CAF-1, it seems likely that the losses of H3.3 in regions that are normally enriched in this variant are a direct consequence of the loss of ASF1, while the gains of H3.3 at regions normally enriched for H3.1 are likely via an indirect compensating mechanism.

## ASF1 is localized to H3.3 enriched genic regions, and its loss affects Pol II occupancy and gene expression

To elucidate the distribution patterns of ASF1 proteins, we performed ChIP-seq using *pASF1A::ASF1A-3xFlag* and *pASF1B::ASF1B-3xFlag* complementing lines (Supplementary Fig. 5a). Consistent with a function in H3.3 deposition, ASF1 proteins were primarily located in the chromosome arms (Supplementary Fig. 5b, c). We also compared ASF1s ChIP signals with RNA polymerase II (Pol II) and other epigenetic features. Correlation analyses of ASF1 patterns with histone variants (H1, H2A.Z, H3.1, and H3.3), histone post-translational modifications, and Pol II revealed that ASF1 proteins exhibit the highest degree of correlation with H3.3, Pol II, and H3K36me2, which is consistent with the role of ASF1s at transcribed regions (Fig. 4a). Next, we examined distribution across gene bodies. ASF1s profile showed a slight dip at the transcriptional start site (TSS) and a narrow peak at the transcriptional termination site (TTS) (Fig. 4b, c). This was similar to the Pol II large subunit (NRPB1) distribution pattern (Fig. 4d, e, and Supplementary Fig. 6a).

To further test the relationship between increased chromatin accessibility and reduced H3.3 occupancy observed in *asf1a1b*, we plotted ASF1s and H3 ChIP-seq signals over the *asf1a1b* more highly accessible regions. ASF1s and H3.3-unique signals showed a strong enrichment over highly accessible regions compared to randomly chosen control regions, suggesting a direct relationship between ASF1 binding to chromatin, H3.3 deposition, and chromatin accessibility (Fig. 4f, g, and Supplementary Fig. 6b). Consistently, NRPB1 binding over highly accessible regions showed a similar pattern (Fig. 4h).

To study the effect of ASF1 proteins on Pol II occupancy, we examined the localization of Pol II in the *asf1a1b* mutant background by ChIP-Seq. Pol II showed a very similar general pattern in wild type and *asf1a1b* except for a slightly elevated level of Pol II occupancy in *asf1a1b* (Fig. 5a). In addition, we observed an increase in Pol II occupancy at *asf1a1b* highly accessible regions (Fig. 5b). To further understand the effects of ASF1 on transcription and how these compared to those of HIRA and CAF-1 complexes, we performed RNA-Seq in wild type, *asf1a1b, fas1-4, fas2-4*, and *hira-1*. The principal component analysis of the expression variation showed that *asf1a1b* grouped with *hira-1*, whereas *fas1-4* and *fas2-4* formed an independent cluster (Fig. 5c, d). These results were confirmed by Pearson correlation with $R = 0.83$, $p < 2.2e−16$, for the *asf1a1b hira-1* pair and $R = 0.93$, $p < 2.2e−16$, for *fas1-4 fas2-4* pair (Supplementary Fig. 6c). Additionally, metaplot analysis of RNA-seq reads over gene bodies revealed an excess towards the 3' end in *asf1a1b* and *hira-1* that was not observed in *fas* mutants (Fig. 5e), which coincided with the localization of ASF1s, NRPB1, and *asf1a1b* highly accessible regions (Fig. 4f–h). We suspected this excess of reads might be due to spurious Pol II transcriptional initiation. To test this, we conducted genome-wide Transcription Start Site Sequencing (TSS-Seq) in *asf1a1b* and wild type[44]. Previous studies suggest that stress can enhance transcription from cryptic promoters[45]. For this reason, we prepared TSS-Seq libraries from samples grown under standard (22 C) and also under heat-stress (37 C) conditions. Compared to wild type, *asf1a1b* exhibited increased levels

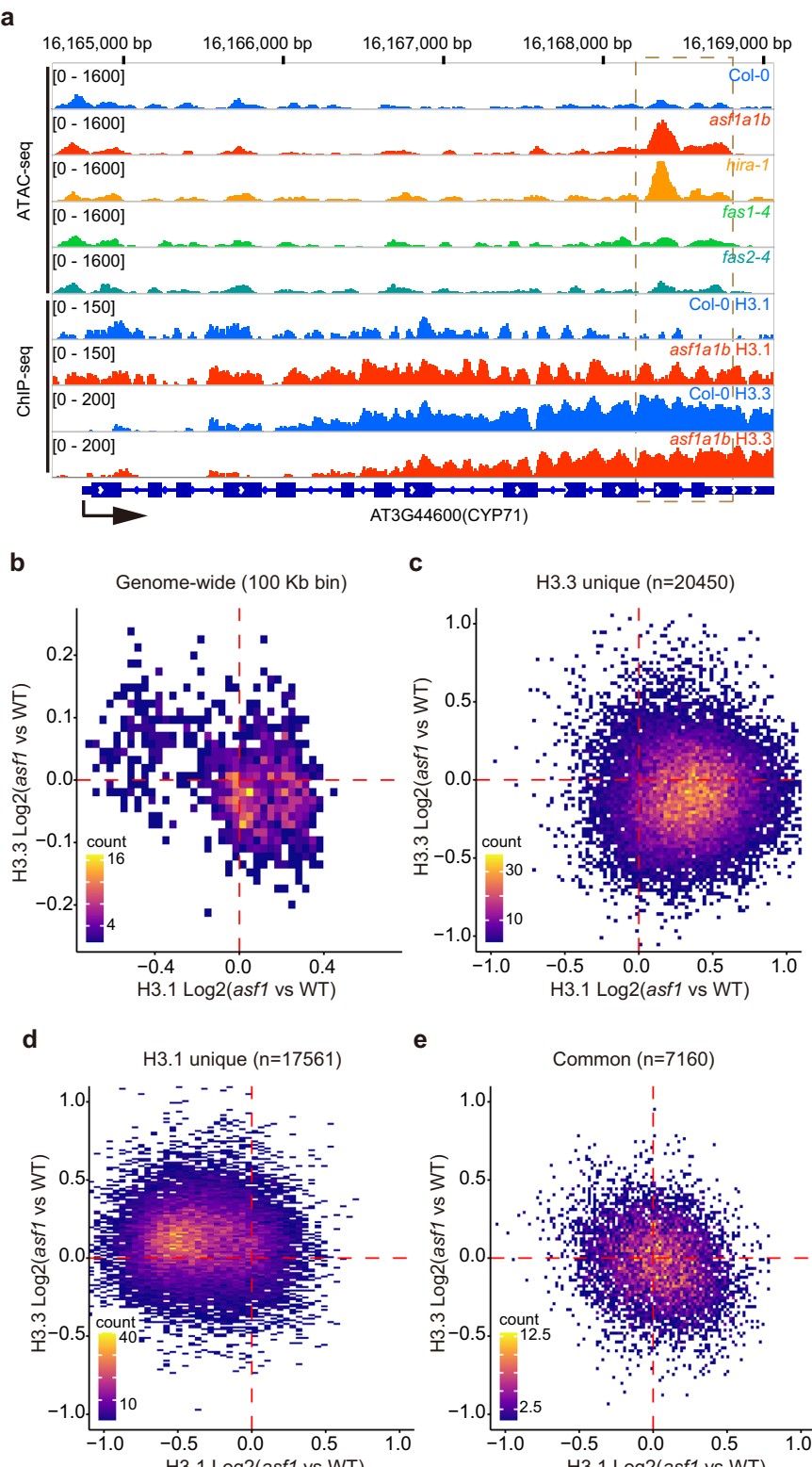

**Fig. 3 | ASF1 mutation causes a redistribution of H3.3 and H3.1. a** Representative example showing H3.1 and H3.3 variation over *asf1a1b* and *hira-1* higher accessible peak. **b** Scatterplot showing correlation of genome-wide H3.1 and H3.3 variation in *asf1a1b* (*asf1a1b* vs. Col-0). The entire genome has been sliced into 100-Kb windows. The colored bin represents the count of dots from low (purple) to high (yellow). **c** Scatterplot showing correlation of H3.1 and H3.3 variation (*asf1a1b* vs. Col-0) over H3.3 unique peaks (*n* = 20450). The colored bin represents the count of dots from low (purple) to high (yellow). **d** Scatterplot showing correlation of H3.1 and H3.3 variation (*asf1a1b* vs. Col-0) over H3.1 unique peaks (*n* = 17561). The colored bin represents the count of dots from low (purple) to high (yellow). **e** Scatterplot showing correlation of H3.1 and H3.3 variation (*asf1a1b* vs. Col-0) over H3.1/H3.3 common peaks (*n* = 7160). The colored bin represents the count of dots from low (purple) to high (yellow).

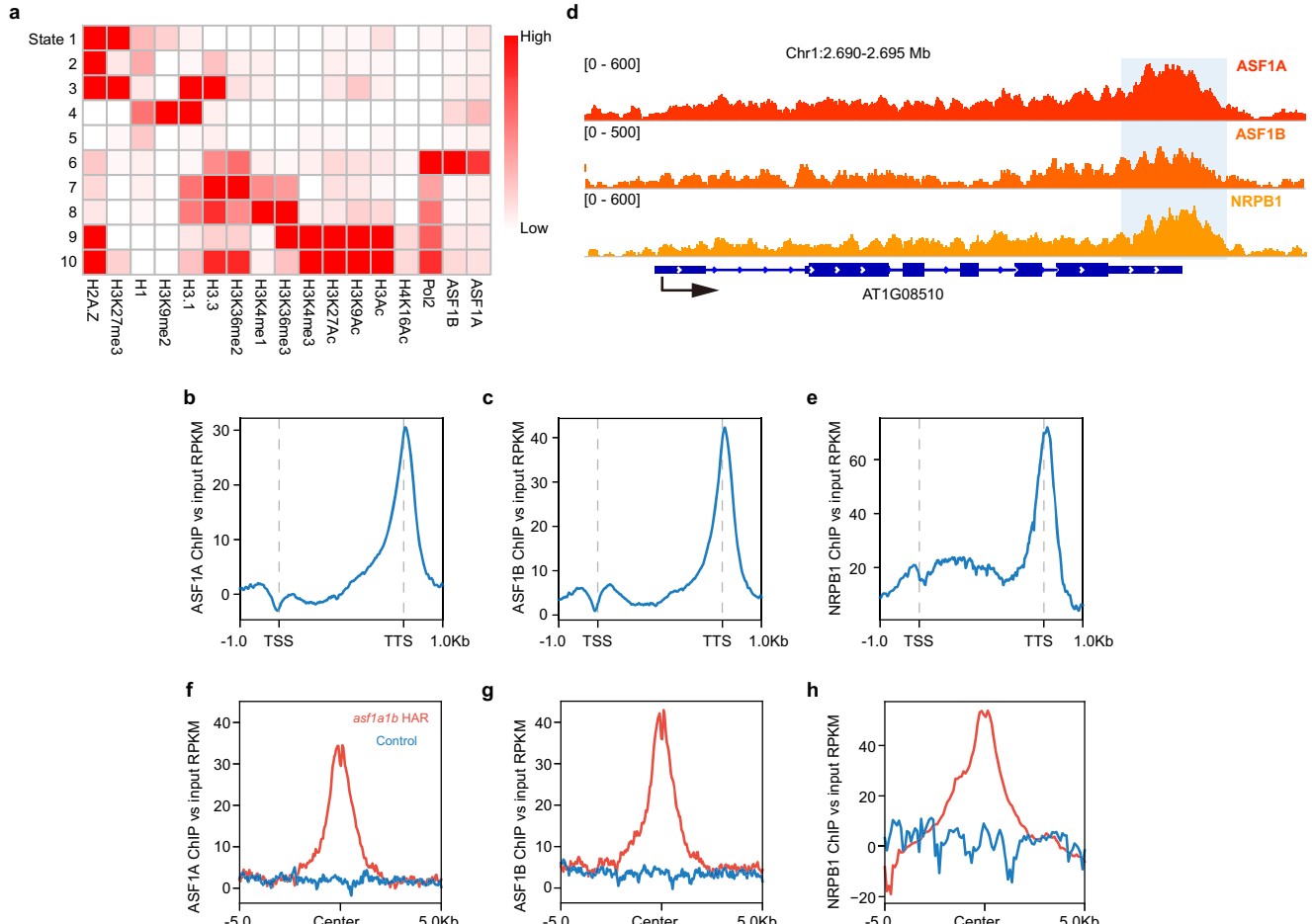

**Fig. 4 | Genomic distribution of ASF1s and NRPB1. a** Overlapping of ASF1 protein distribution with the distribution of histone variants and histone modification marks. Darker red in the heatmap indicates a higher overlapping. State 1 to state 10 represents chromatin environments defined by ChromHMM according to the enrichment of ASF1A/ASF1B and 15 ChIP-seq datasets. **b** Metaplot shows the ASF1A ChIP-seq signal distribution over the gene body, including 1 Kb flanking sequences. **c** Metaplot shows the ASF1B ChIP-seq signal distribution over the gene body, including 1 Kb flanking sequences. **d** Representative example of ChIP-seq tracks of ASF1A, ASF1B, and NRPB1 enriched at TTS. **e** Metaplot showing the distribution of NRPB1 ChIP-seq over gene body with 1 Kb flanking sequences. Metaplot of ASF1A (**f**), ASF1B (**g**), and NRPB1 (**h**) ChIP-seq signal over significantly more accessible peaks of *asf1a1b* (HAR, $n = 1996$). The blue line represents the metaplot of randomly selected control sites.

of cryptic transcription, and this phenotype was enhanced in plants grown under heat stress. The effect was more prominent at highly accessible regions (Fig. 5f, g). These results suggest that ASF1s mediated H3.3 deposition plays a role in preventing spurious Pol II transcriptional initiation.

### *asf1a1b* affects H3.3-associated epigenetic marks

The histone variant H3.3 has been shown to be associated with gene body CG methylation[46] and H3K36me[8], epigenetic marks frequently correlated with gene transcription. To understand how ASF1s affect DNA methylation, we performed whole-genome bisulfite sequencing. We observed that CG methylation over gene bodies was reduced in the *asf1a1b* background (Supplementary Fig. 7a). We also observed a loss of CG methylation when mapping over *asf1a1b* highly accessible regions (Supplementary Fig. 7b). We did the reverse analysis and found that hypomethylated-CG differentially methylated regions (DMRs) in *asf1a1b* corresponded to more open chromatin by ATAC-seq (Supplementary Fig. 7c). Conversely, *asf1a1b* lowly accessible regions showed CG hypermethylation, and hypermethylated-CG DMRs showed less open chromatin by ATAC-seq (Supplementary Fig. 7d, e). These observations are consistent with previous reports showing a relationship between CG methylation and H3.3 density[46] and further

support a role for ASF1s in H3.3 deposition. We also mapped H3K36 di- and tri-methylation using ChIP-Seq, and observed a significant reduction of H3K36me2 at the 3' end of genes in *asf1a1b* compared to wild type; while we observed an increase in H3K36me3 signal at the 5' end and 3' end of genes in *asf1a1b*, suggesting a redistribution of H3K36 methylations in the mutant (Supplementary Fig. 7f, g).

Histone H3.3 is typically depleted at the 5' end of genes and enriched towards the 3' end. This tendency becomes more prominent in medium to highly expressed genes (Supplementary Fig. 7h) and is dependent on gene length[5]. Consistently, we observed that ASF1 proteins were preferentially enriched at the 3' end of the genes, a trend similar to H3.3 distribution, and this became more apparent with increasing gene lengths (Fig. 6a–c). Similarly, some of the alterations observed in *asf1a1b* mutants, such as chromatin accessibility, chromatin density, CG methylation loss, and H3K36me2/3 redistribution, were more prominent with increasing gene length (Fig. 6d–h). Additionally, we observed a similar pattern of DNA methylation alterations in *asf1a1b* and *hira-1*, which was different from that of *fas1-4* and *fas2-4* (Supplementary Fig. 7i–k). In conclusion, these results suggest that proper deposition of H3.3 via ASF1s and HIRA is essential for the maintenance of epigenetic marks and Pol II transcription, especially at longer genes.

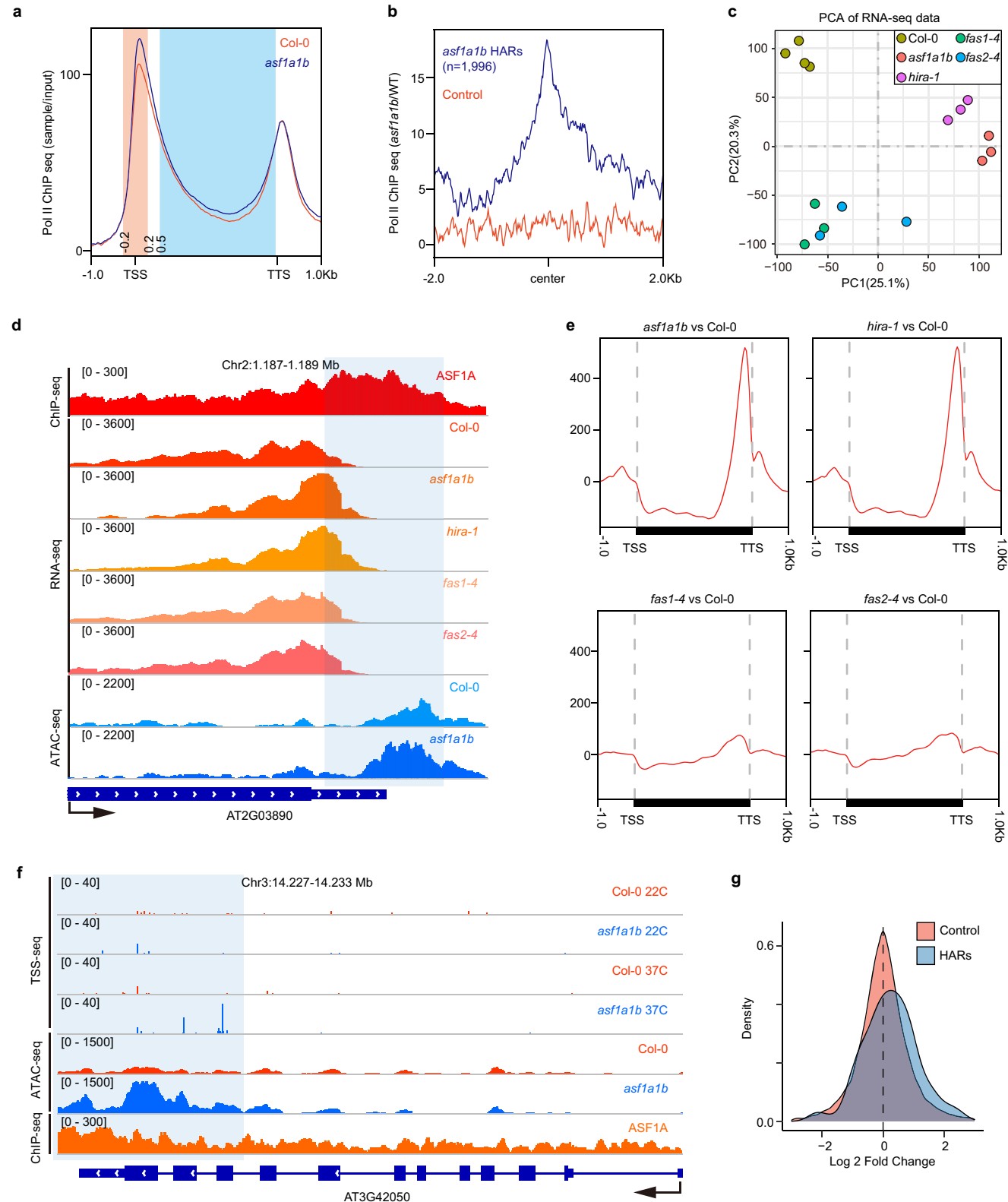

**Fig. 5 | Transcriptomic analysis of H3 Histone chaperone mutants. a** Metaplot showing S2S5-Phospho-Pol II ChIP-seq signal over gene body with 1 Kb flanking sequences. **b** Metaplot of Pol II ChIP-seq signal over *asf1a1b* highly accessible regions (HAR, *n* = 1996). **c** Principal component analysis (PCA) of RNA-seq data of Col-0, *asf1a1b, hira-1, fas1-4,* and *fas2-4*. **d** Representative example of RNA-seq with up-regulation at TTS. **e** Metaplot of RNA-seq over gene body with 1 Kb flanking sequences of *asf1a1b, hira-1, fas1-4,* and *fas2-4* vs. Col-0. **f** Representative example of TSS-seq showing up-regulation at the highly accessible regions at the 3' end of the gene AT3G42050. **g** Fold-change distribution of TSS peaks of *asf1a1b* and wild type at 37 °C compared to control regions.

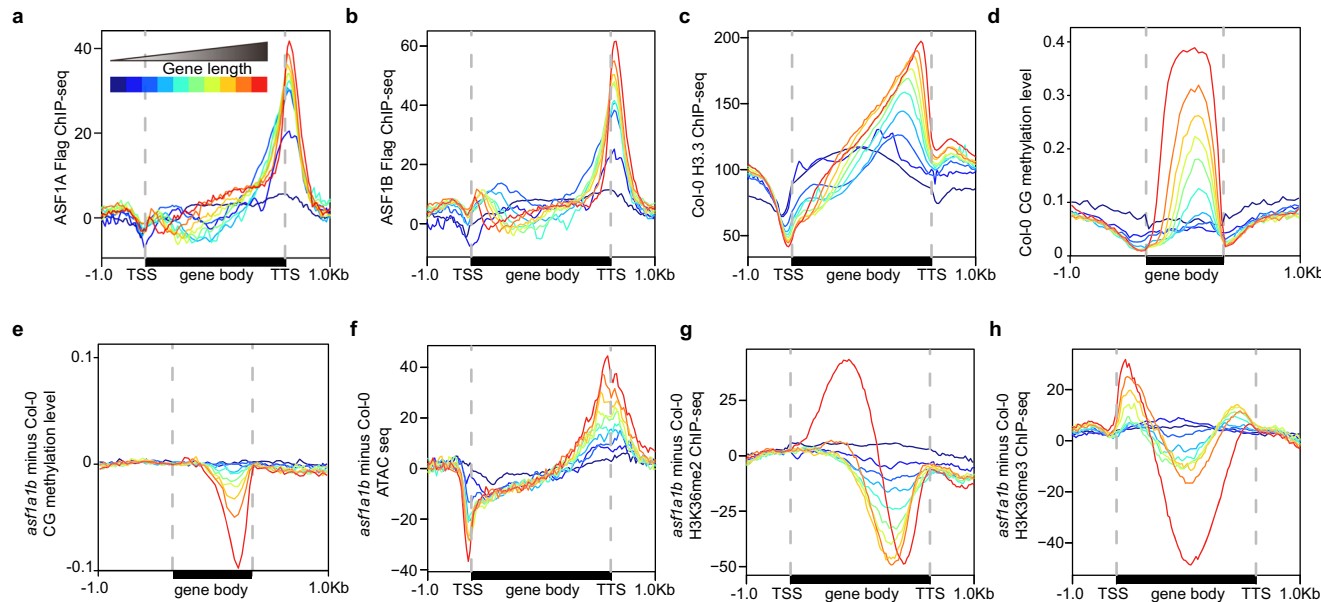

**Fig. 6 | Effects of *asf1a1b* on the distribution of epigenetic marks ranked by gene length. a** Metaplot showing ASF1A ChIP-seq signal over genes grouped by gene length. **b** Metaplot showing ASF1B ChIP-seq signal over genes grouped by gene length. **c** Metaplot showing H3.3 ChIP-seq signal in Col-0 over genes grouped by gene length. **d** Metaplot showing CG methylation levels in Col-0 CG over genes grouped by gene length. **e** Metaplot showing CG methylation level difference in *asf1a1b* vs. Col-0 over genes grouped by gene length. **f** Metaplots showing ATAC-seq signal difference in *asf1a1b* vs. Col-0 over genes grouped by gene length. **g** Metaplots showing H3K36me2 ChIP-seq signal difference in *asf1a1b* vs. Col-0 over genes grouped by gene length. **h** Metaplots showing H3K36me3 ChIP-seq signal difference in *asf1a1b* vs. Col-0 over genes grouped by gene length.

## ASF1s are present in both the cytoplasm and the nucleus

In human cells, Asf1 shuttles between the cytoplasm and the nucleus[29], and in *T. brucei*, Asf1b is mainly located in the nuclei while Asf1a is predominantly cytosolic[47]. We extracted proteins from the cytoplasmic and nuclei-enriched fraction of ASF1-9xMyc-tagged complementing Arabidopsis lines and then performed western blot assays. The results indicated that ASF1s were present in the cytoplasmic and nuclear fractions, suggesting that Arabidopsis ASF1s may also be involved in shuttling histones into the nucleus (Supplementary Fig. 8). This is also consistent with the finding of AtNASP peptides in our IP-MS experiments (Supplementary Table 1).

## Ectopic targeting of ASF1B causes accumulation of both H3 variants at targeted loci

We targeted ASF1B to the nucleosome-depleted promoter region of *FWA*. To do so, we fused ASF1B to the artificial Zinc finger 108 (ZF108) that targets the *FWA* promoter[48,49]. *FWA* expression is controlled by DNA methylation and chromatin accessibility of the tandem repeats present in its promoter[50]. *fwa* epialleles show heritable hypomethylation at the promoter that allows *FWA* ectopic expression, causing late-flowering. On the contrary, re-silencing of *FWA* with ZF108 fusions to components of the RNA-directed DNA methylation system restores flowering time to wild type levels[48,49]. We transformed *pASF1B::ASF1B-ZF108-3xFlag* into the *fwa-4* epiallele and found that it caused a reduction in its flowering time in both the $T_1$ and $T_2$ generations (Fig. 7a, b). Consistent with the flowering time changes, *FWA* expression was significantly reduced in $T_1$ and $T_2$ plants (Fig. 7c, d, Supplementary Fig. 9a, b). This reduction of *FWA* expression was not caused by the re-methylation of the *FWA* tandem repeats (Supplementary Fig. 9c–e). We assayed chromatin compaction by ATAC-Seq and observed that around the ZF108 binding sites, chromatin accessibility was substantially reduced in the *ASF1B-ZF108-3xFlag* line compared to the *fwa-4* control (Fig. 7d).

In addition to *FWA*, ZF108 binds to many off-target sites in the genome[48], which allowed us to examine the effects of ASF1B-ZF108 binding to other loci. We observed that ASF1B-ZF108 had a negative effect on transcription, especially when target sites were within 100 bp of the TSS (Fig. 7e). We hypothesized that the observed reduction in gene expression in lines carrying ASF1B-ZF108 could be caused, in part, by inhibition of transcription factors (TF) binding due to excessive nucleosome density. To test this, we first identified 444 significantly (fold change > 1.2, *p*-value < 0.05) less accessible chromatin peaks within these ZF-ASF1B off-target sites[48] (Fig. 7f). Of these 444 less accessible sites, we focused on 15 sites that were within the promoter regions of genes as defined as minus 1000 bp to zero relative to the TSS. Among these genes, we found that 13 genes were significantly down-regulated (*FWA* plus 12 other genes), one was significantly up-regulated, and the remaining 104 genes were unchanged in expression in ZF-ASF1B (examples shown in Supplementary Fig. 9f–h). We compared the less accessible regions associated with the 13 genes which had decreased expression, with the less accessible regions in the unchanged 104 genes and searched for changes in ATAC-seq signal specifically within the footprint of 572 known transcription factor binding sites found in these regions (TF footprints, see methods). We found that a large number of TF sites showed lower chromatin accessibility in the regions associated with the 13 differentially expressed genes (Fig. 7g). At the same time, there were very few changes at TF sites in genes that did not change in expression (Supplementary Fig. 9i). These results support the hypothesis that the expression alteration observed in ASF1B-ZF108 may be caused in part by the inhibition of TF binding due to excessive histone deposition at promoters.

To test whether the observed increase in chromatin density around ASF1B-ZF108 binding sites was due to an over-accumulation of H3.1 or H3.3, we transformed H3.1 9xMyc or H3.3 9xMyc into ASF1B-ZF108 lines. Interestingly, we observed that the targeting of ASF1B led to an accumulation of both H3.1 and H3.3 to the chromatin at target loci (Fig. 7d, h). Since HIRA-C and CAF1 appear to be at least partially redundant with each other, it is difficult to definitively attribute the observed accumulation of H3 to one of these two complexes[23]. Nonetheless, the results suggest that the tethering of ASF1B can cause, either directly or indirectly, the deposition of both H3.1 and H3.3 to chromatin.

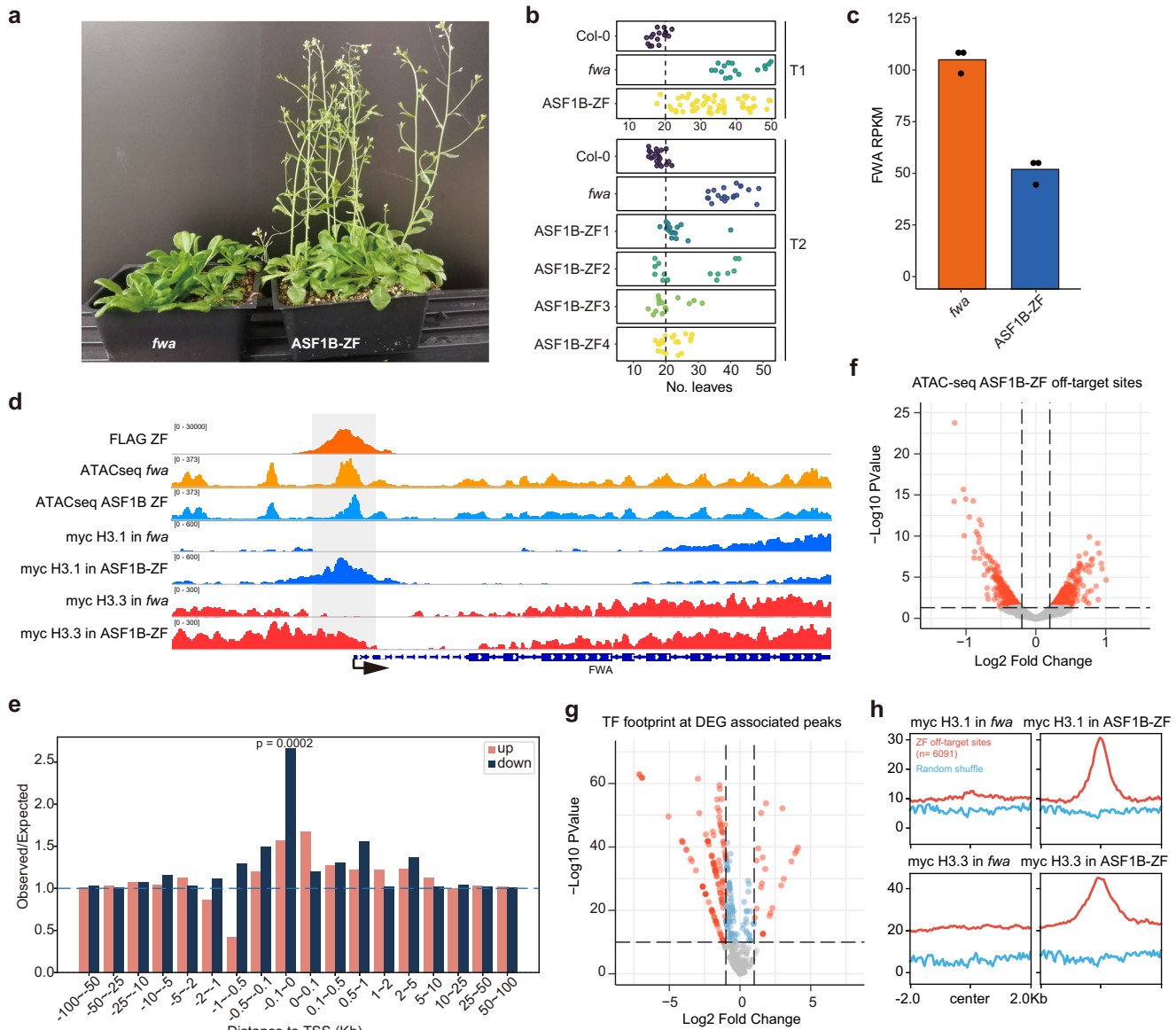

**Fig. 7 | Effects of the targeting of ASF1B to *FWA* promoter. a** Morphological phenotype and flowering time of *fwa-4* and T$_1$ *fwa-4 + ASF1B-ZF108-3xFlag*. **b** The total leaf number of Col-0, *fwa-4*, T$_1$ *fwa-4 + ASF1B-ZF108-3xFlag*, and four T$_2$ *fwa-4 + ASF1B-ZF108-3xFlag* lines. **c** *FWA* expression level in *fwa-4* and T$_1$ *fwa-4 + ASF1B-ZF108-3xFlag*. *n* = 3 biologically independent plants examined. **d** Screenshot showing ASF1B-ZF108-3xFlag binding, chromatin accessibility assayed by ATAC-Seq, and expression level at *FWA*. **e** Observed vs. expected ratio of the association of ASF1B-ZF108-3xFlag binding peaks (*n* = 6091) and DEGs in *ASF1B-ZF108-3xFlag*

lines. *P* value = 0.0002 by hypergeometric test. **f** Volcano plot depicting the differential chromatin accessibility (ATAC-seq) at ZF108 binding peaks (*n* = 6091)[48]. *P* values calculated by two-sided *t* test. **g** The volcano plot shows the differential footprint of 572 plant TF motifs distributed at DEG-associated, down-regulated off-target sites (*n* = 572). **h** Metaplot showing H3.1 or H3.3 ChIP-seq signal over ASF1B-ZF108-3xFlag binding peaks (*n* = 6091) in *fwa-4* or *fwa-4 + ASF1B-ZF108* plants. Blue line represents metaplot of random selected control sites.

## Discussion

ASF1 was initially identified by its ability to depress transcriptional silencing of mating-type loci when overexpressed in yeast[51]. Structural and biochemical evidence showed that mammalian ASF1 binds to H3.1–H4 or H3.3–H4 dimers on one face while leaving H3.1 or H3.3-specific residues exposed[30,31,38]. H3.1 or H3.3-specific residues are recognized by specialized histone chaperone complexes: CAF-1 or HIRA, respectively[30,31,35]. Depleting ASF1 in mammals leads to replication-dependent phenotypes such as an accumulation of cells in S-phase[52,53] and some replication-independent phenotypes[54]. Thus, mammalian ASF1s participate in the replication-dependent H3.1 and the replication-independent H3.3 deposition pathways.

Our proteomic analysis of Arabidopsis ASF1 interacting proteins showed a robust interaction with AtNASP1, HIRA, and all the known

members of the HIRA complex. This suggests that Arabidopsis ASF1s may perform a similar molecular function as its homologs in other organisms, shuttling histone dimers from the cytoplasmic soluble fraction held by NASP and providing them to nuclear complexes. However, whereas yeast and metazoan ASF1 proteins shuttle histones to both HIRA and CAF-1, we did not detect any peptide from any known component of the Arabidopsis CAF-1 complex in our ASF1 spectrometry data. We also examined genetic interactions between *ASF1A/1B* and *FAS2 / HIRA*. The genetic relationship between *ASF1A/1B* and *HIRA* was largely epistatic, given that the triple mutant *asf1a1b hira-1* was hardly distinguishable from *asf1a1b*. By contrast, we could not recover *asf1a1b fas2-4* triple mutants from an F$_2$ population, suggesting lethality and thus a more than additive relationship between ASF1s and FAS2 consistent with them acting in two independent pathways. The

transcriptomic and methylomic analysis also showed that the *asf1a1b* double mutant and *hira-1* single mutant showed a very similar effect on DNA methylation and gene expression, which was different from that of *fas1-4* and *fas2-4*. This is also suggestive of an association between ASF1 and HIRA, and independence from FAS1/2.

We characterized chromatin accessibility in the *asf1a1b* double mutant together with mutants involved in the replication-dependent or replication-independent histone deposition pathways. We found that *asf1a1b* and *hira-1* showed higher chromatin accessibility at the 3' of genes due to a defect in the deposition of H3.3-containing nucleosomes at the affected loci. Consistently, ASF1A and 1B were located at H3.3-rich regions. However, in *asf1a1b*, we also observed a genome-wide redistribution of H3.1 to loci typically occupied by H3.3. A possible explanation is that a reduction in the deposition of H3.3 may result in an ectopic increase of H3.1 deposition by CAF-1. On the other hand, we also observed excessive accumulation of H3.3 at loci normally H3.1-enriched, for instance, at pericentromeric heterochromatin. Since these regions are usually not occupied by ASF1 proteins, it appears that this is an indirect effect. Arabidopsis ATRX is also required to maintain appropriate levels of H3.3 in both the pericentromeric heterochromatin and the chromosome arms[26,27]. Thus, it might be that high levels of unincorporated H3.3 stimulate the activity of ATRX, causing the observed excess of H3.3 at normally H3.1-rich regions. An alternative hypothesis could be that ASF1s could indeed promote the incorporation of H3.1 in heterochromatic regions, but the transitory nature of a putative CAF1-ASF1s interaction would impede its detection by IP-MS. Likewise, if CAF1 recruits ASF1s, the transitory nature of their interactions could also explain why ASF1 is not enriched in our ChIP experiments.

The combinatorial observations that the loss of H3.3 and gain of H3.1 at regions normally enriched in H3.3 were accompanied by over-accumulation of Pol II and increased chromatin accessibility, while the gain of H3.3 and loss of H3.1 at regions normally enriched for H3.1 was accompanied with reduction of chromatin accessibility, supports that ASF1 mediated H3.3 deposition is important for chromatin compaction and is consistent with mammalian studies showing that H3.3 can act as a repressive histone variant[25,55]. While these results seem somewhat counterintuitive, given the finding that H3.3-containing nucleosomes appear to be less stable than those containing H3.1[56], it is likely that not only the identity of the histone variant but also the dynamics of histone deposition by chaperone complexes will dictate the level of chromatin accessibility.

We found that the distribution of ASF1s and Pol II were highly correlated and that regions that become more accessible in *asf1a1b* correspond not only to H3.3-enriched regions but also to regions that would be typically bound by Pol II. Additionally, consistent with genome-wide redistribution of H3.1 and H3.3, epigenetic features known to be associated with H3.3, such as gene body DNA methylation and H3K36me2, were reduced in the *asf1a1b* mutant, whereas H3K36me3, a mark correlated with H3.1, was increased. Plant H3K36me3 has also been considered as a transcription initiation mark[57]. The increase of H3K36me3 together with increased Pol II signal and cryptic transcriptional start sites at the 3' end of gene bodies is highly consistent with this hypothesis.

Although further experiments are still needed to clarify ASF1 function in Arabidopsis, currently available data points to a role mainly in the replication-independent H3 deposition pathway in conjunction with the HIRA complex. ASF1-mediated H3 deposition is crucial for the proper distribution of H3 variants, which in turn impacts the epigenetic landscape regulating transcription in Arabidopsis.

## Methods

### Plant materials and growth conditions
All Arabidopsis plants used in this study are in the Columbia (Col-0) ecotype and were grown at 22 °C in LD (16 h light, 8 h dark) conditions. The Arabidopsis mutants *asf1a-1* (GABI_200G05), *asf1b-1* (SALK_105822C), *fas2-4* (SALK_033228), and *hira-1* (WiscD-sLox362H05) were ordered from Arabidopsis Biological Resource Center. The *fwa-4* epiallele is a line derived from backcrossing the *met1* homozygous mutant with wild type and selecting for progeny that maintained methylation losses at the *FWA* locus[49].

### Epitope-tagged transgenic ASF1A, ASF1B, HIRA, and H3 lines
Full-length genomic DNA fragments, including the native promoter sequences, were cloned into the pENTR/D vector (Invitrogen). The 3xFlag::3C::BLRP, 9xMyc::3C::BLRP, and 3xHA::3C::BLRP tag encoding DNAs were respectively ligated into pENTR/D constructs using AscI (NEB) digestion site in the pENTR/D vector. ASF1A-3xFlag/9xMyc, ASF1B-3xFlag/9xMyc, and HIRA-3HA, H3.1-3xFlag (AT5G10390) and, H3.3-3xFlag (AT4G40040) in-frame fragments were respectively delivered into the binary vector pEG using the LR reaction kit (Invitrogen). Genome-wide distribution of H3.1-3xFlag and H3.3-3xFlag were consistent with previously published ChIP-seq of H3.1/H3.3 fused with 4xMyc-tag[5]. We checked the complementation of the tagged target proteins by rescuing the visible phenotype (for ASF1A and ASF1B constructs) or testing the expression of a downstream gene (for HIRA construct) by RT-qPCR. *asf1a* + ASF1A-3xFlag/9xMyc lines, used for the IP-MS experiment, were obtained by crossing *asf1a* single mutant with *asf1a asf1b* + ASF1A-3xFlag/9xMyc lines. We obtained *asf1b* + ASF1B-3xFlag/9xMyc lines by direct transformation of *asf1b* single mutant after complementation testing in the *asf1a asf1b* double background. Generation of ASF1B-ZF and ASF1A/B-3xflag lines followed the previous approach by cloning ASF1s driven by native promoter into a pEG302 plasmid and transforming the construct into *fwa-4* or *asf1a1b* plants[48]. All primers used in this study are available in Supplementary Table 3.

### IP-MS
About 20 g of young inflorescence tissues were collected from *asf1a* + ASF1A-3xFlag, *asf1b* + ASF1B-9xMyc, and Col-0 (negative control) plants and ground into a fine powder using a RETCH tissue-lyser. We used 500 ul of anti-FLAG M2 magnetic beads (Sigma) or 1000 ul streptavidin beads (Invitrogen) for immunoprecipitation of the target proteins. The detailed procedure for protein extraction and IP-MS is in our previous publications[58,59]. We resuspended dried protein samples in 50 µl digestion buffer (8 M urea, 100 mM Tris-HCl pH 8. Protein disulfide bonds were subjected to reduction using 5 mM Tris (2-carboxyethyl) phosphine for 30 min; free cysteine residues were alkylated by 10 mM iodoacetamide for another 30 min. Samples were diluted with 100 mM Tris-HCl at pH 8 to reach a urea concentration of less than 2 M, then digested sequentially with Lys-C and trypsin at a 1:100 protease-to-peptide ratio for 4 and 12 h, respectively. We terminated the digestion reaction by adding formic acid to 5% (vol/vol) with centrifugation. Finally, samples were desalted using C18 tips (Thermo Scientific, 87784), dried in a SpeedVac vacuum concentrator, and reconstituted in 5% formic acid for LC-MS/MS processing.

We loaded tryptic peptide mixtures onto a 25 cm long, 75 µm inner diameter fused-silica capillary, packed in-house with bulk 1.9 µM ReproSil-Pur beads with 120 Å pores as described previously[1]. Peptides were analyzed using a 140 min water-acetonitrile gradient delivered by a Dionex Ultimate 3000 UHPLC (Thermo Fisher Scientific) operated initially at 400 nL/min flow rate with 1% buffer B (acetonitrile solution with 3% DMSO and 0.1% formic acid) and 99% buffer A (water solution with 3% DMSO and 0.1% formic acid). When we reduced the flow rate to 200 nL/min, we increased buffer B to 6% over 5 min. We applied a linear gradient from 6-28% B to the column over 123 min. The linear gradient of buffer B was further increased to 28-35% for 8 min, followed by a rapid ramp-up to 85% for column washing. We ionized eluted peptides via a Nimbus electrospray ionization source (Phoenix S&T) by applying a distal voltage of 2.2 kV. All label-free mass spectrometry data were collected using data-dependent acquisition on

Orbitrap Fusion Lumos Tribrid mass spectrometer (Thermo Fisher Scientific) with an MS1 resolution of 120,000 followed by sequential MS2 scans at a resolution of 15,000. Database searching was performed with the ProLuCID algorithm against the Arabidopsis proteome database to generate peptide and protein identifications, followed by filtering by DTASelect using a decoy database-estimated false discovery rate of <1%. Precursor/peptide/fragment mass tolerance was 15 ppm. Trypsin digestion was applied for database search allowing a maximum of two missed cleavage sites for arginine and lysine residues at C-terminus.

We defined the interacting proteins as proteins with a fold change (ASF1A or ASF1B's average spectral counts of replicates vs. control's average spectral counts of replicates) >2 and FDR < 0.01 (t-statistics from Linear Models for Microarray Data, LIMMA[60]). Zero or missing values in the control dataset were offset with normalized intensities before transforming to the log-scale to avoid missing values or large variances by LIMMA. The statistical significance of the identified interacting proteins was also determined using SAINT (Source Data)[61].

### Co-immunoprecipitation and Western blot
We used 2 g young inflorescence tissues from each genotype for Co-immunoprecipitation (Co-IP). Anti-HA tag antibodies (ABclonal) were coupled to magnetic beads by Dynabeads antibody coupling kit (Invitrogen) following the description in the manual. We used 1.5 mg antibody-coupled beads for each sample in the Co-IP experiment. In the case of ASF1s-H3s, we used Co-IP anti-FLAG M2 magnetic beads (SIGMA). Described of the detailed procedure is in our previous publication[62]. We used HRP anti-Myc tag antibody (Abcam) with 1:10000 dilution to perform western blots.

### Nuclei extraction and ATAC-seq library preparation
The nuclei collection process from inflorescence and meristem tissues is as described previously[63]. Freshly isolated nuclei were used for ATAC-seq as described elsewhere[41]. We collected inflorescence tissues for extraction of nuclei as follows. About 5 g of inflorescence tissue was collected and immediately transferred into the ice-cold grinding buffer (300 mM sucrose, 20 mM Tris pH 8, 5 mM $MgCl_2$, 5 mM KCl, 0.2% Triton X-100, 5 mM β-mercaptoethanol, and 35% glycerol). The samples were ground with Omni International General Laboratory Homogenizer at 4 °C and then filtered through a two-layer Miracloth and a 40-μm nylon mesh Cell Strainer (Fisher). Samples were spin filtered for 10 min at 3000 g; we discarded the supernatant and then resuspended the pellet with 25 ml of the grinding buffer using a Dounce homogenizer. We performed the wash step twice. We resuspended the nuclei in 0.5 ml of freezing buffer (50 mM Tris pH 8, 5 mM $MgCl_2$, 20% glycerol, and 5 mM β-mercaptoethanol) and then subjected the nuclei to a transposition reaction with Tn5 (Illumina). The transposition reaction was performed with 25 μl of 2x DMF (66 mM Tris-acetate pH 7.8, 132 mM K-Acetate, 20 mM Mg-Acetate, and 32% DMF) was mixed with 2.5 μl Tn5 and 22.5 μl nuclei suspension at 37 °C for 30 min. We purified transposed DNA fragments with the ChIP DNA Clean & Concentrator Kit (Zymo). We prepared libraries with Phusion High-Fidelity DNA Polymerase (NEB) in a system containing 12.5 μl 2x Phusion, 1.25 μl 10 mM Ad1 primer, 1.25 μl 10 mM Ad2 primer, 4 μl ddH2O, and 6 μl purified transposed DNA fragments. We sequenced the ATAC-seq libraries on HiSeq 2500 platform (Illumina).

### Subcellular fractionation
We resuspended 500 mg of ground tissue into 1 mL of Honda buffer (0.44 M Sucrose, 1.25% Ficoll, 2.5% Dextran, 20 mM HEPES pH=7.4, 10 mM MgCl2, 0.5% TritonX-100, 5 mM DTT, 1mMPMSF, 1xPlant Protease Inhibitor -MCE, USA-), incubated on ice for 20 mins occasionally gently inverting the tubes during the incubation. After the incubation, we filtered the suspension through two layers of Miracloth (Millipore, USA). Half of the cleared suspension was saved as input. The rest of the

filtered suspension was spun at 4 °C for 15 min at 2000 × g. The supernatant was saved as non-nuclear proteins. We added TCA to the input and the supernatant (1:5) to precipitate total and non-nuclear proteins, respectively. The resultant pellet was then acetone-washed twice and resuspended in 1xLaemmli buffer (50 mM Tris-HCl pH = 8, 10 mM EDTA, 1% SDS, 1 mM PMSF, and 1xPlant protease inhibitor). We then denatured the mix by boiling it for 15 min. The pellet resulting from the spinning of the original filtered suspension was further washed in Honda buffer four more times or until it looked white, then resuspended in nuclei lysis and incubated on ice for 20 min. The lysate containing nuclear proteins was mixed with Laemmli buffer up to 1x. The mix was then denatured by boiling it for 15 min.

### RNA-seq library preparation
We extracted a plant's total RNA with TRIzol and Direct-zol RNA Miniprep kit (Zymo, R2050) from ~100 mg flower buds. Sequencing libraries were prepared using the TruSeq Stranded mRNA Library Prep kit (Illumina) following the manufacturer's instructions and sequenced on a NovaSeq 6000 sequencer (Illumina).

### ChIP-seq library preparation
We used 10 g of inflorescence and meristem tissues for Flag, HA, Pol II, H3, and H3K36me2/3 ChIP-seq. We performed ChIPs as described previously[64]. Briefly, 2–4 g of flower tissue were collected from 4- to 5-week-old plants and ground with liquid nitrogen. We used a 1% formaldehyde-containing nuclei isolation buffer to fix the chromatin for 10 min. We used fresh-prepared glycine to terminate the crossing reaction, sheared via Bioruptor Plus (Diagenode), and immunoprecipitated with the antibody at 4 °C overnight. Magnetic Protein A and Protein G Dynabeads (Invitrogen) were added and incubated at 4 °C for 2 h. The reverse crosslink was done at 65 °C overnight. The protein-DNA mix was treated with Protease K (Invitrogen) at 45 °C for 4 h. The DNA was purified and precipitated with 3 M Sodium Acetate (Invitrogen), glycoBlue (Invitrogen), and Ethanol at −20 °C overnight. The precipitated DNA was used for library preparation with the Ovation Ultra Low System V2 kit (NuGEN) and sequenced on Illumina NovaSeq. We used the anti-FLAG M2 (Sigma), anti-HA (Roche), anti-Pol II (Ab26721, Abcam, Ser-2 Ser-5 phosphorylated), anti-H3 (Ab1791, Abcam), anti-H3K36me2 (Ab9049, Abcam), and anti-H3K36me3 (Ab9050, Abcam) antibodies in this study. We raised the Anti-NRPB1 antibody in rabbits and further affinity-purified by ABClonal (China) using the peptide HEGDKKDKTGKKDASKDDK. To evaluate Pol II occupancy at both the 5′ end and 3′ end of the gene body, we selected anti-RNA Pol II CTD repeat YSPTSPS antibody, which can capture Pol II signal at both regions (Ser5P and Ser2P). Following the manufacturer's instructions, we prepared libraries with the NuGen Ovation Ultra Low System V2 kit.

### Whole-genome bisulfite sequencing (BS-seq) library preparation
We used leaf tissue as starting material for BS-seq libraries preparation. Genomic DNA was extracted and converted with bisulfite treatment with EpiTect Bisulfite Kit (Qiagen), following the manufacturer's instructions. We sonicated genomic DNA from samples, end-repaired, and ligated with TruSeq DNA single adapters (Illumina) using a Kapa DNA HyperPrep kit (Roche) and converted with an EpiTect Bisulfite Kit (Qiagen). Converted DNA was PCR-amplified by MyTaq polymerase (Bioline) for 12 cycles. The libraries were run on D1000 ScreenTape (Agilent) to determine the quality and size and then purified by AMPure XP beads (Beckman Coulter).

### TSS-seq library preparation
We performed TSS-seq on 14-day-old seedlings grown on MS plates following a previously published protocol[44,65]. We applied heat stress by treating plants for 2 h at 37 °C. 5 μg of total RNA were treated with DNase and CIP (NEB) to remove DNA and all non-capped RNA. Then 5′

caps of capped RNA were removed with Cap-Clip (CellScript). We then ligated single-stranded rP5_RND adapters to 5′-ends with T4 RNA ligase 1 (NEB). Ligated RNAs were enriched and captured by oligo(dT) Dynabeads (Thermo Fisher Scientific). We fragmented Enriched samples for 5 min at 80 °C and generated the first-strand cDNA with SuperScript III (Invitrogen) and random primers. Second-strand cDNA was synthesized with Phusion High-Fidelity DNA Polymerase (NEB) and the BioNotI-P5-PET oligo and captured by Dynabeads for end repairing with End Repair Enzyme Mix (NEB) and ligation with barcoded Illumina compatible adapter using T4 DNA Ligase (NEB). We amplified TSS-seq sequencing libraries and size selected for single-end sequencing with NovaSeq 6000 platform (Illumina).

## BS-PCR and McrBC assay for *FWA* tandem repeat

For BS-PCR of FWA tandem repeat, genomic DNA was extracted from leaf tissue of ASF1B-ZF T2 with a CTAB-based method and converted using the EZ DNA Methylation-Lighting kit (ZYMO research). We amplified the methylated region of the *FWA* promoter region and several control regions with primers described previously[48]. HiSeq 2500 (Illumina) sequencing libraries were made from purified PCR products using a Kapa Hyper DNA Library Prep kit.

For McrBC of *FWA* tandem repeat, genomic DNA was extracted from leaf tissue using a CTAB-based method, treated with PureLink RNase (Invitrogen), and then with McrBC (NEB) for 4 h at 37 °C. We quantified the *FWA* tandem repeat region by qPCR.

## ATAC-seq analysis

We removed ATAC-seq reads adaptors with trim_galore. The reads were then mapped to Arabidopsis thaliana reference genome TAIR10 using Bowtie2 (-X 2000 -m 1)[66]. Reads of chloroplast and mitochondrial DNA were filtered, and we removed duplicate reads using Samtools[67]. We called ATAC-Seq open chromatin peaks of each replicate using MACS2 with parameters -p 0.01 --nomodel --shift −100 --extsize 200. We used Bedtools (v2.26.0) to merge the consensus set of chromatin peaks of each sample intersects, allowing 10 base pairs distance[68]. We used edgeR to define significantly changed peaks [Fold Change, (FC) > 2 and False Discovery Rate, (FDR) < 0.05][69]. Chromosomal distributions of ATAC-seq were calculated by dividing Arabidopsis chromosomes into 100 Kb-sized bins and count reads at each bin with bedtools. We annotated ATAC-seq peak distribution with ChIPseeker[70]. Arabidopsis protein-coding genes were ranked and divided into 10 quantiles according to gene coding region length, and we divided the coding region of each gene in proportion into 10 quantiles. Transcriptional factor footprints were analyzed using TOBIAS[71] with 572 plant TF motifs downloaded from Jasper (http://jaspar.genereg.net/).

## RNA-seq analysis

We aligned cleaned short reads to the reference genome tair10 by Bowtie2 (v2.1.0)[66] and calculated expression abundance using RSEM with default parameters[72]. We visualized with the R package pheatmap[73]. To reduce false-positive of differential expression transcripts with less than five reads of all replicates in total were regarded as lowly expressed genes and have been removed in subsequent analysis. We conducted differential expression analysis using edgeR[69]. We used a threshold of *p*-value <0.05 and Fold Change >2 to decide whether a significant expression difference existed between samples.

## ChIP-seq analysis

We aligned ChIP-seq fastq reads to the TAIR10 reference genome with Bowtie2 (v2.1.0)[66], allowing only uniquely mapping reads with no mismatches. We removed duplicated reads using Samtools. We called ChIP-seq peaks by MACS2 (v2.1.1) and annotated them with ChIPseeker[74]. The bdgdiff function in MACS2 called differential peaks. ChIP-seq data metaplots were plotted by deeptools (v2.5.1)[75]. For Pol II 5′ occupancy analysis, Pol II occupancy was calculated based on

normalized reads count (RPKM) on a TSS +/− 200 bp region and a TSS + 500 bp to TTS gene body region by bedtools. We have listed detailed information for published ChIP-seq datasets in Supplementary Table 4[46,62,76–81]. We defined chromatin states using ChromHMM according to the enrichment of ASF1A/ASF1B and 15 ChIP-seq datasets with numstates 8-19, binsize 50 bp, and foldthresh 1.5[82].

## Whole-genome bisulfite sequencing (BS-seq) analysis

Trim_galore (http://www.bioinformatics.babraham.ac.uk/projects/trim_galore/) was used to trim adapters after filtering low-quality reads. BS-seq reads were aligned to the TAIR10 reference genome by Bismark (v0.18.2)[83] with default settings. Reads with three or more consecutive CHH sites were considered unconverted reads and filtered. DNA methylation levels were defined as #C/ (#C + #T). We called DMRs (Differentially Methylated Regions) using DMRcaller with $p < 0.01$, where the differences in CG, CHG, and CHH methylation were at least 0.4, 0.2, and 0.1, respectively[84].

## TSS-seq analysis

We analyzed TSS-seq data following previously a published pipeline[44]. We trimmed TSS-seq reads with Trim_galore and 5′ end UMI barcodes using UMI-Tools[85]. The reads were then aligned to TAIR10 genome assembly using STAR (v2.7.0e)[86]. We filtered mapping files with MAPQ < 10 and deduplicated using SAMtools (v1.9)[67] before being converted to stranded Bedgraph files using bedtools (v2.26.0)[68]. We identified TSS peaks with CAGEfightR (v0.99.0)[87], and differential TSS peaks [Fold Change, (FC) > 2 and *p*-value < 0.05] were called with DESeq2 (v1.28.1)[88].

## BS-PCR analysis

We analyzed BS-PCR data with a previously published pipeline[48]. BS-PCR data were trimmed with primer sequences and mapped to TAIR10 reference genome with bsmap (v2.90), allowing two mismatches and one best hit (-v 2 -w 1)[89].

## Reporting summary

Further information on research design is available in the Nature Portfolio Reporting Summary linked to this article.

# Data availability

Data supporting the findings of this work are available within the paper and its Supplementary Information files. All high-throughput sequencing data generated in this study are accessible at NCBI's Gene Expression Omnibus (GEO) via GEO Series accession number GSE188493. We have deposited the mass spectrometry proteomics data to the ProteomeXchange Consortium via the PRIDE partner repository with the dataset identifier PXD035578. Source data are provided with this paper.

# Code availability

The customized codes used in this study are available upon reasonable request to the corresponding author.

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

## Acknowledgements

We thank members of Jacobsen and Ausin laboratories for helpful discussion. We thank Dr. Jason Gardiner for editing the manuscript. We are also grateful to M. Akhavan and the UCLA BSCRC High Throughput BioSequencing Core for technical assistance. The work in the I.A. laboratory was supported by the National Science Foundation of China (grant numbers: 31870270 and 31801025). The work in the Jacobsen laboratory was supported by NIH grant R35 GM130272. S.E.J. is an investigator of the Howard Hughes Medical Institute.

## Author contributions

Conceptualization, Z.Z., I.A., and S.J.; project supervision, I.A., S.J.; experiments, Z.Z., Y.W., M.W., assisted by F.Y., Q.T., Y.X., Y.Z. and L.X.; data analysis, Z.Z., W.L.; mass spectrometry & data analysis, Y.J.A., and J.A.W.; high throughput sequencing, S.F.; manuscript draft, Z.Z., I.A. and S.J. with help from S.F. and S.M.; funding acquisition, I.A., S.J. All authors approved the final version of the manuscript and agreed on the content and conclusions.

## Competing interests

The authors declare no competing interests.
