## [Peer Review File · Nature Communications]

Histone chaperone ASF1 mediates H3.3-H4 deposition in ArabidopsisReviewer #1 (Remarks to the Author):

This manuscript by Zhong et al., investigated the function of Arabidopsis ASF1 in histone deposition and chromatin regulation. Overall this work has been well performed and provided convincing data to show that ASF1 regulates H3.3 deposition and the epigenome. Please see my concerns and suggestions below.

1. For the protein interaction data, were H3.3 peptides (but not H3.1 peptides) being identified in ASF1a/b IP-MS experiments? The Ala/Thr amino acid at position 31 could help to distinguish H3.1/H3.3 in protein MS. Besides the ASF1-HIRA co-IP experiment, another co-IP experiment to test the interaction between ASF1 and H3.1/H3.3 could be helpful. This is because ASF1 may directly interact with H3.1 independent of CAF1.
2. H3.1 and H3.3 ChIP-seq experiments were performed in Col, fas, asf1 and hira. It needs to be stated clearly whether native H3.1/H3.3 antibodies were used or tag lines (and by crossing them into fas, asf1 and hira mutants) were used in these experiments. If native antibodies were used, evidence proving their specificity to Arabidopsis H3.1 and H3.3 needs to be provided, and the H3.1 and H3.3 genome distribution profiles need to be compared with previous data generated with tag lines.
3. Similar to the reduction of H3.3 deposition in asf1, H3.1 enrichment was also reduced in asf1 especially at TE and H3.1 unique peaks. Could it rather indicate that ASF1 promotes H3.1 deposition but not due to an indirect mechanism? The ASF1-CAF1 interaction could be more transient than that of ASF1-HIRA, this may explain why ASF1-CAF1 interaction was not detected and that ASF1 is absent from H3.1 enriched regions (e.g. heterochromatin).
4. A previous study on asf1 mutant has shown increased S-phase arrest and DNA damage and reduced endoreduplication in asf1 (Zhu et al., Plant Journal), indicating that ASF1 may play a role in chromatin assembly during DNA replication. These results could be discussed in the manuscript.
5. The ectopic targeting of ASF1 could be a nice experiment to test its specificity. Although current results showed that H3 was enriched at ASF1 ectopically targeted regions, H3.1 and H3.3 ChIP-seq will help to further reveal which H3 type is deposited.

Reviewer #2 (Remarks to the Author):

The manuscript of Zhong et al. describes an interesting demonstration that the protein ASF1 is used by Arabidopsis to load the histone dimer H3.3-H4 on the HIRA complex for translocating it into the nucleus. I anticipate that I am no expert in Arabidopsis and the majority of the techniques described in the paper. However, I think the project is well executed.

My analysis of the manuscript is focused on the IP experiment performed with mass spectrometry. I appreciate the author providing a table with comprehensive data points for the proteins quantified in the different samples/replicates. However, I would recommend the following:

- 1) The authors should provide more insights regarding the IP experiment analyzed via mass spectrometry, including data analysis. The authors cite this paper for details (PMID: 31350403). However, this other paper does not include any information regarding which setup was used, chromatographic gradient, type of mass spectrometer, acquisition method, software and software settings for peptide identification.

2) Related to my previous question, how is this "normalized spectral abundance" calculated? In my experience, it is hard to see such a clean output from an IP experiment. I do not doubt the authenticity of the results, but it would be helpful to know how these numbers are calculated.

In fact, the authors should represent this data as volcano plot, including statistical assessment of the enrichment. For instance, is UBN2 significant in the ASF1A IP? It is very variable.

3) How do these proteins rank in abundance compared to other proteins identified in the experiment? For instance, are there histones in the list provided by the MS experiment? That would be a good positive control. It would be even better if the authors could define unique peptides for the histone variant H3.3 from the spectra; it will help validating their claims.

I do not have much to add on the rest. In agreement with the Editor, I have focused my review on the IP experiment. Before I recommend publication, I would like to see more details on the experimental design and the result table provided by MS.

Reviewer #3 (Remarks to the Author):

In this manuscript, the authors examine the role of Anti-Silencing Factor 1 (ASF1) in the deposition of histone H3 variants in plants. Using genomic and proteomic analyses of Arabidopsis wild type (Col-0) and mutant lines (asf, fas, and hira), they show preference of ASF1 proteins for the H3.3-H4 deposition via HIRA pathway. This study provides new knowledge about the function of plant histone chaperones. However, in my opinion, both genomic and mainly proteomic results should be better described and further supported by supplementary data to be suitable for publishing in Nature Communications.

I have the following suggestions to improve the manuscript:

1) I can't entirely agree that fas1 mutant shows mild developmental abnormalities as stated in Line 87: "The fact that fas1-4 and hira-1 mutations do not shown dramatic developmental abnormalities..."

Previously, substantial developmental abnormalities of fas1 mutant have been reported:

- The fas1 mutant shows fasciated stems, dentate leaves, short roots, small siliques with a decreased number of seeds connected with fertility problems (Kaya et al., 2001; Exner et al., 2006; Mozgova et al., 2010; Varas et al., 2015).**
- fas1-4 displays 3% of FAS1 mRNA expression levels and the truncated protein generated is unable to interact with PCNA and other subunits of the CAF-1 complex (Ramirez-Parra and Gutierrez, 2007a). It appears to be the strongest allele described to date since it exhibits the most severe developmental phenotype and ~96-fold more intrachromosomal HR events than WT (Kaya et al., 2001; Kirik et al., 2006).**

2) Col-0 used as a control for IP-MS is not appropriate. Respective plant lines with empty tag vectors would be more suitable to identify and filter out non-specifically bound proteins to tag. Similarly, the signal of myc on WB in Co-IP experiment can be detected due to non-specific binding.

3) No details are provided about LC-MS/MS analysis, database search, and data evaluation. The authors have to enclose supporting data for IP-MS results. Which other

proteins were significantly enriched next to proteins presented in Table 1?
Table 1 is confusing, including the title and caption. The table doesn't show "Summary of peptides" but "Identified ASF1-interacting partners". What does it mean "Normalized spectral abundance factor value"?

I would suggest simplifying the table as follows:

- Remove Col-0 from the table as no identifications are there, and add info about control into the caption

- Present quantitative data as median values from all replicates

- If the protein was not identified, use "N/F" instead of "0".

The authors show the abundance of interacting partners of ASF1-A and ASF1B proteins but did not comment within the text. What was the aim - to show differences in the levels of ASF1-A and ASF1B binding proteins or to prove their presence?

4) What does it mean "State 1-10" in Fig. 3a?

5) Line 313: "The results indicated that ASF1s were present in the cytoplasmic and nuclear fractions at comparable levels, suggesting that Arabidopsis ASF1s may also be involved in shuttling histones into the nucleus (Figure S6)."

Noticeably, WB detected a much stronger signal in the cytoplasmic fraction (Figure S6). Input for WB should be shown. What are all the signals on WB? Why is there a shift in MW of detected bands between cytoplasmic and nuclear fractions on WB? How did the authors perform fractionation of plant material into cytoplasmic and nuclei-enriched fractions? I'm surprised that so pure cytoplasmic and nuclear fractions were obtained (as evident from anti-H3 signal).

6) Overall, the manuscript's weakness is an insufficient method description. All genomic and proteomic methods are described poorly, often referencing previously published studies where another reference appears instead of method description (e.g., CHIP-seq, see ref 60). The authors should refer to original articles and avoid inappropriate self-citations. The paragraph "Plasmid Construction" at line 448 is empty.

Given that the methods used are not trivial, I recommend enclosing all detailed descriptions into supplementary material. While high-throughput sequencing data are accessible, raw proteomic data not. The authors have to deposit also proteomic data into a public depository.

7) The manuscript should be revised for English language (typing errors, syntax):

Line 66: change "...4-5 amino-acid" for "...4-5 amino-acids"

Line 87: change "do not shown" for "do not show"

Line 113: change "differential" for "different" or "distinct"

Line 119: rephrase "...double mutant significantly inhibits plant growth..."

Line 132: change "...distribution histone..." for "...distribution of histone..."

Line 137: change "...which play critical roles epigenetic..." for "...which plays critical roles in epigenetic..."

Lines 142-144: rephrase "The CAF-1 and HIRA deposition pathways important for deposition of their corresponding histone H3 variant at specific regions of the genome, which would explain the developmental abnormalities of fas and hira-1 mutants."

Line 465: change "is available" for "are available"

Line 418: rephrase "...it is likely that not only the identify of the histone variant, but also the dynamics of histone deposition by chaperone complexes will dictate..."

...etc.

Reviewer #4 (Remarks to the Author):

The work of Zhong et al. focuses on investigating the role of histone chaperone ASF1 in H3 deposition. The authors present a broad spectrum of genetic data (crosses of various mutants), genomic (ATACseq, ChIPseq, bisulfate sequencing) and proteomic (CoIP, IP-MS) data to show that, unlike in yeast and metazoan where ASF1 cooperates with CAF-1 and HIRA complexes in H3 deposition, Arabidopsis ASF1 strongly prefers HIRA. In this way ASF1 contributes to the deposition of the H3.3 variant. Consequently, mutations in ASF1 result in redistribution of H3.3 and H3.1. More importantly, the loss of ASF1 significantly affects the changes in RNAPII occupancy. The authors also showed that ASF1 plays a role in preventing spurious RNAPII transcriptional initiation. As the cherry on top, the authors showed that targeting ARF1 to hypomethylated FWA causes its re-silencing, leading to a reduction in flowering-time. This targeting was carried out by ZF108, which has many additional targets in the genome. Therefore, the authors using this system could also investigate behavior of other genes and suggested that ZF-ASF1, by binding to their promoters, can affect their transcription by limiting the binding of TFs (but see my specific comments below).

Although the changes observed by the authors are not dramatic, it is due to the key role of H3 in chromatin, the associated redundancy of the pathways of its incorporation and partially overlapping functions of different H3 variants. The number of research approaches and results (especially large-scale data) presented in the work is huge and allows the authors for multi-level understanding of the examined processes of H3 variant deposition and their consequences on gene transcription. All the work is written in a very logical and transparent manner. Therefore, I am convinced that it should be accepted by Nature Communications.

Below are some comments that, in my opinion, could further strengthen the conclusions and improve the reception of the work:

1. In the experiment with ASF1 targeting by ZF, the authors showed that there is no change in DNA methylation (as exemplified by FWA). Instead, the authors observed changes in the chromatin compaction (via ATAC-Seq) and concluded that the decrease in expression was due to a reduction in the binding of transcription factors. But won't the ZF binding itself block the TF binding site or prevent the formation of RNAPII preinitiation complex via a steric hindrance? The authors should include in the experiment shown in figure 6a-6c also a control in which *fwa-4* is transformed with ZF108 without fused ASF1.
2. The authors present two results showing that ASF1 shuttles histones to HIRA but not to CAF-1: IP-MS and genetic interactions. IP-MS did not return any components of the CAF-1 complex. But as absence of evidence is not evidence of absence, the genetic confirmation is important. Not obtaining a triple mutant (*asf1a1b fas2*) in F2 generation does not have to mean synthetic lethality, but can be simply a result of segregation bias. The authors could select the F2 plant with one of the mutation segregating, self-fertilize it and then screen the F3 population (or show siliques with lacking seeds corresponding to triple mutants).
3. It would be worth presenting a more detailed validation of the phenotype similarity between the *asf1a1b hira-1* triple mutant and the *asf1a1b* double mutant. This would allow a more precise determination of the functional relationships between ASF1 and HIRA.
4. As the two variants differ in their chromosomal distribution (H3.3 along gene-rich chromosome arms, H3.1 overrepresented in pericentromeric regions) would it be possible to compare them between wild type and *asf1a1b* at the scale of whole chromosomes? In addition, the authors could further discuss alternative H3 deposition pathways in the discussion.

Minor comments:

1. Line 142 - the sentence is missing a verb.

- 2. Fig. 3 - please, characterize chromatin states 1-10 as shown in Figure 3a.**
- 3. HAR and LAR should be explained in S2f and S2g**

We thank all reviewers for their positive comments and constructive suggestions. We have revised our manuscript according to the reviewers' recommendations and provided our point-to-point response below **with responses in blue bold type**.

REVIEWER COMMENTS

Reviewer #1 (Remarks to the Author):

This manuscript by Zhong et al., investigated the function of Arabidopsis ASF1 in histone deposition and chromatin regulation. Overall this work has been well performed and provided convincing data to show that ASF1 regulates H3.3 deposition and the epigenome. Please see my concerns and suggestions below.

We thank the reviewer for these positive comments.

1. For the protein interaction data, were H3.3 peptides (but not H3.1 peptides) being identified in ASF1a/b IP-MS experiments? The Ala/Thr amino acid at position 31 could help to distinguish H3.1/H3.3 in protein MS. Besides the ASF1-HIRA co-IP experiment, another co-IP experiment to test the interaction between ASF1 and H3.1/H3.3 could be helpful. This is because ASF1 may directly interact with H3.1 independent of CAF1.

Thank you for the suggestion. We have reanalyzed peptides identified by ASF1A/B IP-MS and specifically examined amino acids that can distinguish H3.1/H3.3. Although the peptide count was relatively low, both H3.1 and H3.3 peptides were identified in the ASF1A/B IP-MS experiments. In the case of ASF1A-3XFlag, 3, 4, and 4 H3.1 peptides were detected (in replicate 1, 2, and 3, respectively) and 4, 5, and 3 peptides of H3.3 were detected in replicate 1, 2, and 3, respectively. In the case of ASF1B-9xMyc, we also see peptides from both H3.1 and H3.3. However, the WT control also yielded peptides for both H3.1 and H3.3.

We also performed Co-IP assays to test whether ASF1 proteins interact with H3.1 and H3.3 and found that both ASF1 proteins can interact with H3.1 and H3.3. However, since H3.1 and H3.3 interact with AtNASP, which also interacts with both ASF1 proteins, this interaction might be explained regardless of any interaction with the HIRA complex or CAF1. These results can be found in supplementary figure s2i and below.

We thus have added a subsection titled 'ASF1 proteins physically interact with H3.1 and H3.3 in vivo' to the main text L198~L206.

Figure S2i: Co-immunoprecipitation assay of ASF1A, ASF1B, H3.1 and H3.3. The antibody used for the western blot is indicated to the right.

2. H3.1 and H3.3 ChIP-seq experiments were performed in Col, fas, asf1 and hira. It needs to be

stated clearly whether native H3.1/H3.3 antibodies were used or tag lines (and by crossing them into *fas*, *asf1* and *hira* mutants) were used in these experiments. If native antibodies were used, evidence proving their specificity to Arabidopsis H3.1 and H3.3 needs to be provided, and the H3.1 and H3.3 genome distribution profiles need to be compared with previous data generated with tag lines.

Thank you for the comment. We have used H3.1 (At5g10390) and H3.3 (At4g40040) driven by their promoters and tagged with Myc or Flag tags. Genome-wide distribution of H3.1-3xFlag and H3.3-3xFlag were consistent with previously published ChIP-seq with 4xMyc-tag [Stroud 2012 PNAS]. This is now included in the methods L483~L486.

3. Similar to the reduction of H3.3 deposition in *asf1*, H3.1 enrichment was also reduced in *asf1* especially at TE and H3.1 unique peaks. Could it rather indicate that ASF1 promotes H3.1 deposition but not due to an indirect mechanism? The ASF1-CAF1 interaction could be more transient than that of ASF1-HIRA, this may explain why ASF1-CAF1 interaction was not detected and that ASF1 is absent from H3.1 enriched regions (e.g. heterochromatin).

We appreciate the suggestion; Given the currently available data, this is still a possibility that has to be considered. We have now added to the discussion the following sentences (L428~L432):

“...An alternative hypothesis could be that ASF1s could indeed promote the incorporation of H3.1 in heterochromatic regions, but the transience of a putative CAF1-ASF1s interaction would impede its detection by IP-MS. Likewise, if CAF1 recruits ASF1s, the transitory nature of their interactions could also explain why ASF1s are not enriched in our ChIP experiments...”

4. A previous study on *asf1* mutant has shown increased S-phase arrest and DNA damage and reduced endoreduplication in *asf1* (Zhu et al., Plant Journal), indicating that ASF1 may play a role in chromatin assembly during DNA replication. These results could be discussed in the manuscript.

Thank you for the suggestion. The function of ASF1 in chromatin assembly during DNA replication is an important paper in plant ASF1 research. We have added this reference to the introduction (L119~L120).

5. The ectopic targeting of ASF1 could be a nice experiment to test its specificity. Although current results showed that H3 was enriched at ASF1 ectopically targeted regions, H3.1 and H3.3 ChIP-seq will help to further reveal which H3 type is deposited.

Thank you for the suggestion. We now have performed ChIP-Seq of H3.1-Myc and H3.3-Myc in the ASF1B-ZF108 background. These experiments showed that ASF1B-ZF108 co-localized with H3.1 and H3.3 at the *FWA* promoter and the ZF108 off-targets, suggesting that at least in this artificial system, targeting ASF1B to the chromatin promotes the ectopic deposition of both H3.1 and H3.3. Since CAF1 and HIRA are likely partially redundant, they can likely deposit both histone variants meaning that we cannot definitively determine which complex is mediating the observed ectopic deposition of histones in the ASF1B tethering system. These new results (Figure 7h, L371-379) and discussion have been added to the results section.

Reviewer #2 (Remarks to the Author):

The manuscript of Zhong et al. describes an interesting demonstration that the protein ASF1 is

used by Arabidopsis to load the histone dimer H3.3-H4 on the HIRA complex for translocating it into the nucleus. I anticipate that I am no expert in Arabidopsis and the majority of the techniques described in the paper. However, I think the project is well executed.

We thank the reviewer for these positive comments.

My analysis of the manuscript is focused on the IP experiment performed with mass spectrometry. I appreciate the author providing a table with comprehensive data points for the proteins quantified in the different samples/replicates. However, I would recommend the following:

1) The authors should provide more insights regarding the IP experiment analyzed via mass spectrometry, including data analysis. The authors cite this paper for details (PMID: 31350403). However, this other paper does not include any information regarding which setup was used, chromatographic gradient, type of mass spectrometer, acquisition method, software, and software settings for peptide identification.

We are happy to provide this, and a more detailed description of IP-MS is now in the methods. Please see method section L501-L531 for more detailed descriptions.

2) Related to my previous question, how is this “normalized spectral abundance” calculated? In my experience, it is hard to see such a clean output from an IP experiment. I do not doubt the authenticity of the results, but it would be helpful to know how these numbers are calculated.

In fact, the authors should represent this data as volcano plot, including statistical assessment of the enrichment. For instance, is UBN2 significant in the ASF1A IP? It is very variable.

Thank you for the suggestion. The IP-MS results represent the number of the normalized spectral abundance factor multiplied by 10e5 (10e5*NSAF). The following formula was used to calculate the number of the normalized spectral abundance factor.

$$(\text{NSAF})_J = \frac{(\text{Sc}/L)_J}{\sum_{i=1}^N (\text{Sc}/L)_i}$$

**Sc: spectral counts
L: Protein length**

Since the number is presented as 10e5*NSAF, the number looks cleaner.

As suggested by the reviewer, we also included statistical assessment and volcano plots of the proteins identified (Figures 2a and b, and shown below). We used FDR < 0.01 and FC > 2 as cutoff. UBN2 in ASF1A IP-MS assays is very variable; its FDR = 0.91, but its FDR = 0.0003 in ASF1B, so we conclude this protein should remain in the table.

Figure 2a-b: Volcano plots showing proteins interact with ASF1A (a) and ASF1B (b) identified by IP-MS. The distribution of identified proteins was plotted according to the Log2 fold change of normalized spectral abundance factor multiplied by 10e5 (10e5*NSAF) and $-\text{Log}_{10}$ FDR values. Proteins with FDR < 0.01 were labeled in red.

3) How do these proteins rank in abundance compared to other proteins identified in the experiment? For instance, are there histones in the list provided by the MS experiment? That would be a good positive control. It would be even better if the authors could define unique peptides for the histone variant H3.3 from the spectra; it will help validating their claims.

We have re-analyzed the IP-MS data to see whether it's possible to get the unique peptides that can separate H3.1 and H3.3. Both H3.1 and H3.3 unique peptides can be identified in ASF1A and ASF1B (see below). The peptide count is very low, however, and we have not included this. Reviewer1 also asked us to do co-IP with H3.1 and H3.3, and we also found that ASF1 interacted with both, a result that has been added to the paper. The interpretation of the specificity for HIRA has been addressed in the revised manuscript.

We identified H3.1 in ASF1A:

We identified H3.3 in ASF1A:

We identified H3.3 in ASF1B:

We identified H3.1 in ASF1B:

However, we also have identified H3.1 and H3.3 in Col-0:

I do not have much to add on the rest. In agreement with the Editor, I have focused my review on the IP experiment. Before I recommend publication, I would like to see more details on the experimental design and the result table provided by MS.

Thank you again for helping us providing more details on the IP-Mass spec aspects of the paper.

Reviewer #3 (Remarks to the Author):

In this manuscript, the authors examine the role of Anti-Silencing Factor 1 (ASF1) in the deposition of histone H3 variants in plants. Using genomic and proteomic analyses of Arabidopsis wild type (Col-0) and mutant lines (*asf*, *fas*, and *hira*), they show preference of ASF1 proteins for the H3.3-H4 deposition via HIRA pathway. This study provides new knowledge about the function of plant histone chaperones. However, in my opinion, both genomic and mainly proteomic results should be better described and further supported by supplementary data to be suitable for publishing in Nature Communications.

I have the following suggestions to improve the manuscript:

1) I can't entirely agree that *fas1* mutant shows mild developmental abnormalities as stated in Line 87: "The fact that *fas1-4* and *hira-1* mutations do not show dramatic developmental abnormalities..."

Previously, substantial developmental abnormalities of *fas1* mutant have been reported:

- The *fas1* mutant shows fasciated stems, dentate leaves, short roots, small siliques with a decreased number of seeds connected with fertility problems (Kaya et al., 2001; Exner et al., 2006; Mozgova et al., 2010; Varas et al., 2015).
- *fas1-4* displays 3% of FAS1 mRNA expression levels and the truncated protein generated is unable to interact with PCNA and other subunits of the CAF-1 complex (Ramirez-Parra and Gutierrez, 2007a). It appears to be the strongest allele described to date since it exhibits the

most severe developmental phenotype and ~96-fold more intrachromosomal HR events than WT (Kaya et al., 2001; Kirik et al., 2006).

Thank you for correcting our imprecise description of *fas1*. We simply were impressed with the fact that the plant is still alive. We certainly didn't mean to hide or give a false impression. Indeed, in Figure 2 we show *fas2-4* and *fas1-4* phenotypes grown in our conditions in which developmental abnormalities are indeed visible. We mainly wanted to point out that the phenotype is not as dramatic as expected if H3.1 could not be deposited after every round of replication in this mutant. In addition, the actual visual phenotype of *fas* mutants could be somewhat misleading since it can be mainly rescued by downregulating the salicylic acid signaling pathway, indicating that most of this phenotype could be an effect of a constitutively active SA pathway. Please see our changes at Line 75~78 and Line 88~90.

We have corrected these sentences and changed them to:

"...Although fas mutants exhibit several morphological and molecular defects, they produce viable progeny¹⁴ and incorporation of H3.1 is only partially abolished^{15,16}. Indeed, most of the fas2-4 syndrome is rescued by downregulating the Salicylic acid (SA) pathway, indicating that the pleiotropism might be mostly an effect of constitutive activation of the SA pathway rather than a massive loss of H3.1 deposition¹⁷."

"...The fact that fas1-4 hira-1 double mutants display a more severe phenotype than the single mutants suggests some degree of redundancy between the CAF-1 and the HIRA pathways..."

2) Col-0 used as a control for IP-MS is not appropriate. Respective plant lines with empty tag vectors would be more suitable to identify and filter out non-specifically bound proteins to tag. Similarly, the signal of myc on WB in Co-IP experiment can be detected due to non-specific binding.

Thank you for the comment. We agree that an empty vector line showing a similar expression level as our tagged genes may have been a more appropriate control. However, there are a few reasons why we believe the detected interactions are real: i) ASF1A and ASF1B IPs were performed using two different methods (α Flag IP+Flag peptide release for 1A and Streptavidin IP+3C release for 1B) and in both cases, similar proteins are being enriched (i.e., HIRA complex members). ii) The Co-IP assay was also performed in the opposite direction; HIRA was tagged with 3xHA and IPed using an α HA antibody, and we can detect bands that are the same size as our Myc-tagged ASF1s proteins in the input. iii) We have performed (and made publicly available) Flag IPs and Streptavidin IPs in many papers, and have never detected a significant enrichment of any of the HIRA complex members or NASP or TOUSLED. We hope the reviewer is convinced by these arguments and we are happy to add anything else to the paper if it is helpful.

3) No details are provided about LC-MS/MS analysis, database search, and data evaluation. The authors have to enclose supporting data for IP-MS results. Which other proteins were significantly enriched next to proteins presented in Table 1?

Reviewer 2 also asked for more detail and present the data as volcano plots. We have added much more detailed information about LC-MS/MS method and analysis in the methods section (L501~531). We have also included a statistical assessment and volcano plots of the protein identified, as shown below and in Figures 2a and b. As added in new method part L534-536, we used FDR < 0.01 and FC > 2 as cutoff.

Table 1 is confusing, including the title and caption. The table doesn't show "Summary of peptides" but "Identified ASF1-interacting partners". What does it mean "Normalized spectral abundance factor value"?

Normalized spectral abundance factor value ($10e5 \cdot \text{NSAF}$). It's calculated by

$$(\text{NSAF})_J = \frac{(\text{Sc}/L)_J}{\sum_{i=1}^N (\text{Sc}/L)_i}$$

Sc: spectral counts

L: Protein length

I would suggest simplifying the table as follows:

- Remove Col-0 from the table as no identifications are there, and add info about control into the caption

Thanks for the suggestion. We have removed Col-0 from the table and revised the caption into 'Supplementary Table 1 Identified ASF1-interacting proteins by immunoprecipitation in combination with mass spectrometry (IP-MS) in ASF1A-3xFlag and ASF1B-9xMyc lines. Normalized spectral abundance factor value ($10e5 \cdot \text{NSAF}$) values for each protein are shown. Corresponding 4 wild type control for ASF1A and 3 wild type control for ASF1B didn't show significant enrichment of peptides'.

- Present quantitative data as median values from all replicates

To show the robustness of the data, we think it may be better to show all replicates, so we didn't use median values.

- If the protein was not identified, use "N/F" instead of "0".

We have changed "0" into "N/F" and added "N/F represents not identified" into the caption (L1164).

The authors show the abundance of interacting partners of ASF1-A and ASF1B proteins but did not comment within the text. What was the aim - to show differences in the levels of ASF1-A and ASF1B binding proteins or to prove their presence?

We showed the abundance of interacting partners of ASF 1A and ASF1B proteins to prove that 1) they share very similar interacting partners. 2) ASF1A and ASF1B don't interact with each other. The following sentences were used to comment on this result (L175-L184) "In both cases, we identified AtNASP, HIRA, and all the known members of the HIRA complex, including CABIN1, UBN1, and UBN2 (Supplementary Table1, Figure 2a and 2b)"; "Although the two ASF1 proteins shared most of the interacting partners, we did not observe physical interaction between them, suggesting that in Arabidopsis, two mutually exclusive HIRA complexes may exist depending on whether they include ASF1A or ASF1B".

4) What does it mean "State 1-10" in Fig. 3a?

Chromatin states were defined by ChromHMM according to the enrichment of ASF1A/ASF1B and 15 ChIP-seq datasets. States 1-10 are 10 states of the chromatin regions typically enriched with different epigenetic marks that are more or less associated. For example, State 1 Includes regions marked by the simultaneous presence of H2A.Z and H3K27me3; State 6 is defined by enrichment of H3.3, H3K36me2, Pol 2, and ASF1 proteins. We have added the meaning of State1-10 at figure legend of Figure 4a (L995-996) and method part (Line689~691).

5) Line 313: "The results indicated that ASF1s was present in the cytoplasmic and nuclear fractions at comparable levels, suggesting that Arabidopsis ASF1s may also be involved in shuttling histones into the nucleus (Figure S6)."

Noticeably, WB detected a much stronger signal in the cytoplasmic fraction (Figure S6). Input for WB should be shown. What are all the signals on WB? Why is there a shift in MW of detected

bands between cytoplasmic and nuclear fractions on WB? How did the authors perform fractionation of plant material into cytoplasmic and nuclei-enriched fractions? I'm surprised that so pure cytoplasmic and nuclear fractions were obtained (as evident from anti-H3 signal).

Thank you for the comments.

We agree that using the word comparable was not correct, and we have removed this so that the sentence now reads:

“...The results indicated that ASF1s were present in the cytoplasmic and nuclear fractions, suggesting that in Arabidopsis ASF1s may also be involved in shuttling histones into the nucleus...”

Different banding patterns and shifts are observed for ASF1 proteins and we assume this depends on the solubility of the modified versions in different extraction buffers plus the exposure time used to develop the WB. We don't know what the post-translational modifications of the ASF1 proteins are. Since we don't have anything definitive to say, we have not commented on this in the manuscript.

We apologize for the lack of clarity on the protocol. We have enriched nuclei following the Honda Buffer protocol, which has been added in detail to the methods section under the epigraph “Subcellular fractionation”

6) Overall, the manuscript's weakness is an insufficient method description. All genomic and proteomic methods are described poorly, often referencing previously published studies where another reference appears instead of method description (e.g., CHIP-seq, see ref 60). The authors should refer to original articles and avoid inappropriate self-citations. The paragraph "Plasmid Construction" at line 448 is empty.

Given that the methods used are not trivial, I recommend enclosing all detailed descriptions into supplementary material. While high-throughput sequencing data are accessible, raw proteomic data not. The authors have to deposit also proteomic data into a public depository.

We apologize for not providing enough details. We have thoroughly revised the methods to make the protocols more detailed. For example, detailed descriptions of IP-MS are available at L501~531, the Subcellular fractionation method available at line 566~583, ChIP-seq at L593~604, and WGBS at L617~622. We have also sent the proteomic data to the ProteomeXchange Consortium database under accession number px-submission #585873. The empty paragraph "Plasmid Construction" at line 448 has been removed.

7) The manuscript should be revised for English language (typing errors, syntax):
Line 66: change "...4-5 amino-acid" for "...4-5 amino-acids"

Revised into 4~5 amino acids.

Line 87: change "do not shown" for "do not show"

Revised into “The fact that fas1-4 hira-1 double mutants display a more severe phenotype than the single mutants suggests some degree of redundancy between the CAF-1 and the HIRA pathways”.

Line 113: change "differential" for "different" or "distinct"

Revised into “exhibit distinct expression patterns”.

Line 119: rephrase "...double mutant significantly inhibits plant growth..."

Revised into “Depletion of either ASF1A or ASF1B does not cause apparent morphological defects, whereas the simultaneous mutation of ASF1A and ASF1B

significantly inhibits plant growth, affects reproductive organ development, alters the response to heat stress, and affects chromatin assembly during replication”

Line 132: change "...distribution histone..." for "...distribution of histone..."

Revised into “distribution of histone post-translation modifications”.

Line 137: change "...which play critical roles epigenetic..." for "...which plays critical roles in epigenetic..."

Revised into “ASF1s are essential in supplying histone variants to the HIRA complex, which plays critical roles in epigenetic gene regulation”.

Lines 142-144: rephrase "The CAF-1 and HIRA deposition pathways important for deposition of their corresponding histone H3 variant at specific regions of the genome, which would explain the developmental abnormalities of fas and hira-1 mutants."

Revised into “The CAF-1 and HIRA deposition pathways are essential for the deposition of their corresponding histone H3 variants at specific regions of the genome”.

Line 465: change "is available" for "are available"

Revised into “are available”.

Line 418: rephrase "...it is likely that not only the identify of the histone variant, but also the dynamics of histone deposition by chaperone complexes will dictate..."
...etc.

Revised into “it is likely that not only the identity of the histone variant”.

We apologize for the mistakes. We have now corrected these issues as suggested by the reviewer and gone through the manuscript for other language issues.

Reviewer #4 (Remarks to the Author):

The work of Zhong et al. focuses on investigating the role of histone chaperone ASF1 in H3 deposition. The authors present a broad spectrum of genetic data (crosses of various mutants), genomic (ATACseq, ChIPseq, bisulfate sequencing) and proteomic (CoIP, IP-MS) data to show that, unlike in yeast and metazoan where ASF1 cooperates with CAF-1 and HIRA complexes in H3 deposition, Arabidopsis ASF1 strongly prefers HIRA. In this way ASF1 contributes to the deposition of the H3.3 variant. Consequently, mutations in ASF1 result in redistribution of H3.3 and H3.1. More importantly, the loss of ASF1 significantly affects the changes in RNAPII occupancy. The authors also showed that ASF1 plays a role in preventing spurious RNAPII transcriptional initiation. As the cherry on top, the authors showed that targeting ARF1 to hypomethylated FWA causes its re-silencing, leading to a reduction in flowering-time. This targeting was carried out by ZF108, which has many additional targets in the genome. Therefore, the authors using this system could also investigate behavior of other genes and suggested that ZF-ASF1, by binding to their promoters, can affect their transcription by limiting the binding of TFs (but see my specific comments below).

Although the changes observed by the authors are not dramatic, it is due to the key role of H3 in chromatin, the associated redundancy of the pathways of its incorporation and partially overlapping functions of different H3 variants. The number of research approaches and results (especially large-scale data) presented in the work is huge and allows the authors for multi-level understanding of the examined processes of H3 variant deposition and their consequences on gene transcription. All the work is written in a very logical and transparent manner. Therefore, I am convinced that it should be accepted by Nature Communications.

Below are some comments that, in my opinion, could further strengthen the conclusions and improve the reception of the work:

Thank you for these enthusiastic comments!

1. In the experiment with ASF1 targeting by ZF, the authors showed that there is no change in DNA methylation (as exemplified by FWA). Instead, the authors observed changes in the chromatin compaction (via ATAC-Seq) and concluded that the decrease in expression was due to a reduction in the binding of transcription factors. But won't the ZF binding itself block the TF binding site or prevent the formation of RNAPII preinitiation complex via a steric hindrance? The authors should include in the experiment shown in figure 6a-6c also a control in which *fwa-4* is transformed with ZF108 without fused ASF1.

Thank you for the comment.

We have not used “ZF108 only” as a control. However, we have fused the same ZF108 to several different proteins and a YPet fluorescent protein, but we didn't observe a negative effect on *FWA* or other off-target sites' transcriptional repression in our previously published paper (Gallego-Bartolome, 2019 Cell), suggesting ZF binding will not block the TF binding site. In addition, as shown at Fig. S7i, many peaks have ZF binding, but TF bindings are not affected.

2. The authors present two results showing that ASF1 shuttles histones to HIRA but not to CAF-1: IP-MS and genetic interactions. IP-MS did not return any components of the CAF-1 complex. But as absence of evidence is not evidence of absence, the genetic confirmation is important. Not obtaining a triple mutant (*asf1a1b fas2*) in F2 generation does not have to mean synthetic lethality, but can be simply a result of segregation bias. The authors could select the F2 plant with one of the mutation segregating, self-fertilize it and then screen the F3 population (or show siliques with lacking seeds corresponding to triple mutants).

We appreciate this suggestion. To address this, we collected seeds from *asf1a-1 asf1b-1 FAS2/fas2-4* as well as *asf1a-1 asf1b-1 FAS1/fas1-4* individuals and genotyped the F3 population. As expected, we did not find any homozygous *fas2-4* or *fas1-4*; instead, we observed a nearly 1:1 ratio of *FAS2/fas2-4* or *FAS1/fas1-4*, suggesting that the triple mutant combination blocks the development of one of the gametophytes. We have now added Supplementary table 2 showing the segregation ratios.

Supplementary Table 2 Segregation ratios of *asf1a-1 asf1b-1 FAS1/fas1-4* and *asf1a-1 asf1b-1 FAS2/fas2-4* genotyped at F3 population.

asf1a1b fas1-4 (het)	Expected	Observed	asf1a1b fas2-4 (het)	Expected	Observed
FAS1	23.5	48	FAS2	18	37
fas1-4	23.5	0	fas2-4	18	0
het	47	46	het	36	35

3. It would be worth presenting a more detailed validation of the phenotype similarity between the *asf1a1b hira-1* triple mutant and the *asf1a1b* double mutant. This would allow a more precise determination of the functional relationships between ASF1 and HIRA.

Thank you for the suggestion. We have now added extra pictures taken at 28, 35, and 42 days after germination illustrating the phenotypes of *asf1a1b* double, *asf1a1b hira-1* triple, and their respective controls (please see Supplementary figure 2).

4. As the two variants differ in their chromosomal distribution (H3.3 along gene-rich chromosome arms, H3.1 overrepresented in pericentromeric regions) would it be possible to compare them

between wild type and *asf1a1b* at the scale of whole chromosomes? In addition, the authors could further discuss alternative H3 deposition pathways in the discussion.

This is an excellent suggestion. A comparison of the distribution of H3.1 and H3.3 in the wild type versus *asf1a1b* double mutant background was already shown in Supplementary Figure 2l (see below). H3.1 and H3.3 are normally enriched at heterochromatin and euchromatin respectively. The loss of ASF1s leads to an increase of H3.1 in euchromatin and a decrease at heterochromatin. While H3.3 showed the opposite pattern. We observed an increase of H3.3 at heterochromatin and a decrease at euchromatin. Alternative H3 deposition pathways are discussed at L432-439 in the discussion part.

Figure S2l: Metaplot of H3.1 and H3.3 variation (*asf1a1b* vs Col-0) over 5 chromosomes splitting into 100 Kb bins. The Blue and red lines represent H3.1 and H3.3 variation, respectively.

Minor comments:

1. Line 142 - the sentence is missing a verb.

We revised this sentence into “The CAF-1 and HIRA deposition pathways are essential for the deposition of their corresponding histone H3 variants at specific regions of the genome”.

2. Fig. 3 - please, characterize chromatin states 1-10 as shown in Figure 3a.

States 1-10 are 10 states of the chromatin regions typically enriched with different epigenetic marks that are more or less associated. For example, State 1 Includes regions marked by the simultaneous presence of H2A.Z and H3K27me3; State 6 is defined by enrichment of H3.3, H3K36me2, Pol 2, and ASF1 proteins. We have added the meaning of State1-10 at figure legend of Figure 4a (previously Fig. 3a) and method part Line689~691.

3. HAR and LAR should be explained in S2f and S2g

Corrected as suggested. We have revised S2f and S2g legends into “(f) Genomic distribution of *asf1a1b*, *hira-1*, *fas1-4*, and *fas2-4*. Highly Accessible Regions (HARs) represent genomic regions where mutants have higher chromatin accessibility than wild-type. (g) Genomic distribution of *asf1a1b*, *hira-1*, *fas1-4*, and *fas2-4*. Lowly Accessible Regions (LARs) represent genomic regions where mutants have lower chromatin accessibility than wild-type ones. Enrichment = (number of sample peaks)/(number of randomly selected peaks)”.

Reviewer #1 (Remarks to the Author):

The authors have adequately addressed my comments.

Reviewer #2 (Remarks to the Author):

I am satisfied with the responses to my questions provided by the authors. From my perspective, this article is worthy of publication in Nature Communications.

Reviewer #3 (Remarks to the Author):

I appreciated the authors' replying to my comments and the correction of several issues. However, I am still not convinced that this manuscript is suitable for publication in such a prestigious journal as Nature Communications, as many points have not yet been addressed.

1) Previously, I asked the authors to enclose supporting data for IP-MS results including a list of all proteins identified in the IP-MS experiment. The authors added volcano plots to the manuscript (Fig. 2a and 2b) but the supporting data including protein IDs are still missing. It is incredible that only the HIRA pathway proteins were found to be significant in the IP-MS experiment. I'd like to ask the authors once again to present the list of all identified proteins together with a detailed calculation of values presented in volcano plots.

The description of MS analysis, database search and data evaluation (lines 523 – 531) is insufficient. For instance, there is no information about precursors isolation and fragmentation. Parameters of database search are missing (enzyme specificity, number of missed cleavages, modifications, mass error tolerance, ...). The data evaluation is described as follows: "We defined the interacting proteins as proteins with a fold change of normalized spectral abundance factor multiplied by 10^5 ($10^5 \cdot \text{NSAF}$) > 2 ." It's unclear to me how fold changes of NSAF values were calculated from the replicates and how significantly enriched proteins were determined. I would really appreciate an excel file with all calculations to be added to the supporting material.

2) The formula for calculating NSAF should be added in Methods with a description of all variables. What exactly represents the denominator? Description of "i" and "l" is missing.

3) The authors added a description of the fractionation of plant material into cytoplasmic and nuclei-enriched fractions to the Methods. If I caught it well (lines 580 - 582), they analyzed chromatin-bound proteins instead of the whole nuclear extract by WB. Why was the supernatant discarded? The authors should comment on it in the text and precisely specify the type of fraction in the text (line 322) and Figure S6 caption.

4) The authors uploaded the proteomic data to the ProteomeXchange Consortium database but haven't provided the account details to the reviewers (username and password). Without access to the data, the reviewers cannot consider the relevance of proteomic data.

5) In Line 195 the authors stated: "IP-MS experiments did not yield a significant number of peptides from H3.1 and H3.3, possibly due to the low solubility of histones in neutral pH buffers."

That's an entirely odd argument. Histones are soluble in neutral pH buffers. I suppose that they occur in the neutral pH environment even in vivo. Further, I suppose that co-immunoprecipitation of H3 variants with ASF1A and ASF1B proteins has also been performed in a neutral pH buffer. The majority of histone peptides were probably lost during chromatography (as tryptic peptides are too short and hydrophilic). In addition, longer post-translationally modified histone peptides were probably lost during database searches. The authors can repeat the database search with acetylation and methylation to be set as variable modifications and with a higher number of missed cleavages.

6) In Line 395 the authors stated: "Our proteomic analysis of Arabidopsis ASF1 interacting

proteins showed a robust interaction with AtNASP1, HIRA, and all the known members of the HIRA complex."

Line 399: "However, whereas yeast and metazoan ASF1 proteins shuttle histones to both HIRA and CAF-1, we did not detect any peptide from any known component of the Arabidopsis CAF-1 complex in our ASF1 spectrometry data."

At this point, I have to stress once again that proteomic data supporting the authors' claims have not been shown.

Reviewer #4 (Remarks to the Author):

The revised manuscript has been much improved. The authors have addressed most of my concerns. However, I could not find pictures of asf1a1b double, asf1a1b hira-1 triple, and their respective controls, which according to the authors had been included in Figure S2 as a response to my suggestion #3. It seems to me that the authors did not include them in the revision by mistake.

We would like to thank all four reviewers for these additional helpful comments. We have addressed each in our point-by-point response below.

Reviewer #1 (Remarks to the Author):

The authors have adequately addressed my comments.

Response:

Thank you again for reviewing our manuscript.

Reviewer #2 (Remarks to the Author):

I am satisfied with the responses to my questions provided by the authors. From my perspective, this article is worthy of publication in Nature Communications.

Response:

Thank you again for reviewing our manuscript.

Reviewer #3 (Remarks to the Author):

I appreciated the authors' replying to my comments and the correction of several issues. However, I am still not convinced that this manuscript is suitable for publication in such a prestigious journal as Nature Communications, as many points have not yet been addressed.

Response:

Thank you again for reviewing our manuscript, and we apologize for not sufficiently addressing the comments in the first round.

1) Previously, I asked the authors to enclose supporting data for IP-MS results including a list of all proteins identified in the IP-MS experiment. The authors added volcano plots to the manuscript (Fig. 2a and 2b) but the supporting data including protein IDs are still missing. It is incredible that only the HIRA pathway proteins were found to be significant in the IP-MS experiment.

I'd like to ask the authors once again to present the list of all identified proteins together with a detailed calculation of values presented in volcano plots.

Response:

Thank you for this suggestion. We have added all identified proteins and corresponding statistics presented in the volcano plots into the source data. Please see tabs labeled Fig.2a and Fig.2b (the proteins are ranked by the FDR value < 0.01 and fold change > 2). We got ~2000 protein hits in our IP-MS, from which only HIRA complex components were significantly enriched.

The description of MS analysis, database search and data evaluation (lines 523 – 531) is insufficient. For instance, there is no information about precursors isolation and fragmentation. Parameters of database search are missing (enzyme specificity, number of missed cleavages, modifications, mass error tolerance, ...). The data evaluation is described as follows: "We defined the interacting proteins as proteins with a fold change of normalized spectral abundance factor multiplied by 10e5 ($10e5 * \text{NSAF}$) > 2." It's unclear to me how fold changes of NSAF values were calculated from the replicates and how significantly enriched proteins were determined.

I would really appreciate an excel file with all calculations to be added to the supporting material.

Response:

We apologize for the lack of detail. We have followed very common protocols for the IP-MS experiments, which have been used in dozens of papers from our lab. To be more clear about this description, we have added the following details in the methods section: "Precursor/peptide/fragment mass tolerance was 15 ppm. Trypsin digestion was applied for database search allowing a maximum of two missed cleavage sites for arginine and lysine residues at C-terminus. The number of the normalized spectral abundance factor (NSAF) was calculated as below. The number of MS spectra matching protein J (S_c) was normalized by the protein length (L), and divided by the total number of S_c/L for all proteins (from index i to N) identified in IP-MS.

$$(\text{NSAF})_J = \frac{\left(\frac{S_c}{L}\right)_J}{\sum_{i=1}^N \left(\frac{S_c}{L}\right)_i}$$

S_c : Spectral counts

L: Protein length

We defined the interacting proteins as proteins with a fold change (ASF1A or ASF1B's average spectral counts [we used spectral counts rather than NSAF. We apologize for this error in our previous rebuttal.] of replicates vs. control's average spectral counts of replicates) > 2 and FDR < 0.01 (t-statistics from Linear Models for Microarray Data, LIMMA). Zero or missing values in the control dataset were offset with normalized intensities before transforming to the log-scale to avoid missing values or large variances by LIMMA. The statistical

significance of the identified interacting proteins was also determined using SAINT (provided in Source data).”

We submitted raw data as source data (excel spreadsheet for Fig. 2a and Fig. 2b), which contains fold change and FDR calculations of the IP-MS. We cross-validated this result with the SAINT algorithm, which gives similar results (detailed results included in Source data).

2) The formula for calculating NSAF should be added in Methods, describing all variables. What exactly represents the denominator? Description of “i” and “l” is missing.

Response:

J stands for protein J, and i stands for index (from 1 to N). Since they are basic indexes in different formulas, we didn’t add them in our previous revision. We added the following descriptions in the method.

$$(\text{NSAF})_J = \frac{\left(\frac{\text{Sc}}{L}\right)_J}{\sum_{i=1}^N \left(\frac{\text{Sc}}{L}\right)_i}$$

Sc: Spectral counts

L: Protein length

The number of the normalized spectral abundance factor (NSAF) was calculated as below. The number of MS spectra matching protein J (Sc) was normalized by the protein length (L), and divided by the total number of Sc/L for all proteins (from index i to N) identified in IP-MS.

3) The authors added a description of the fractionation of plant material into cytoplasmic and nuclei-enriched fractions to the Methods. If I caught it well (lines 580 - 582), they analyzed chromatin-bound proteins instead of the whole nuclear extract by WB. Why was the supernatant discarded? The authors should comment on it in the text and precisely specify the type of fraction in the text (line 322) and Figure S6 caption.

Response:

We are sorry about the mistake here; we did not discard the lysate after nuclei lysis buffer, indeed we did not centrifuge the lysate but precipitated it instead. We rewrote this part as follows:

“We resuspended 500 mg of ground tissue into 1mL of Honda buffer (0.44M Sucrose, 1.25% Ficoll, 2.5% Dextran, 20mM HEPES pH=7.4, 10mM MgCl₂, 0.5%

TritonX-100, 5mM DTT, 1mMPMSF, 1xPlant Protease Inhibitor -MCE, USA-), incubated on ice for 20 mins occasionally gently inverting the tubes during the incubation. After the incubation, we filtered the suspension through two layers of Miracloth (Millipore, USA). Half of the cleared suspension was saved as input. The rest of the filtered suspension was spun at 4°C for 15 minutes at 2000g. The supernatant was saved as non-nuclear proteins. We added TCA to the input and the supernatant (1:5) to precipitate total and non-nuclear proteins, respectively. The resultant pellet was then acetone-washed twice and resuspended in 1xLaemmli buffer (50mM Tris-HCl pH=8, 10mM EDTA, 1% SDS, 1mM PMSF, and 1xPlant protease inhibitor). We then denatured the mix by boiling it for 15 min. The pellet resulting from the spinning of the original filtered suspension was further washed in Honda buffer four more times or until it looked white, then resuspended in nuclei lysis and incubated on ice for 20 min. The lysate containing nuclear proteins was mixed with Laemmli buffer up to 1x. The mix was then denatured by boiling it for 15 min.”

4) The authors uploaded the proteomic data to the ProteomeXchange Consortium database but haven't provided the account details to the reviewers (username and password). Without access to the data, the reviewers cannot consider the relevance of proteomic data.

Response:

We apologize for the inconvenience. PRIDE and GEO account details have been provided in the Nature Reporting Summary.

5) In Line 195 the authors stated: "IP-MS experiments did not yield a significant number of peptides from H3.1 and H3.3, possibly due to the low solubility of histones in neutral pH buffers."

That's an entirely odd argument. Histones are soluble in neutral pH buffers. I suppose that they occur in the neutral pH environment even in vivo. Further, I suppose that co-immunoprecipitation of H3 variants with ASF1A and ASF1B proteins has also been performed in a neutral pH buffer.

The majority of histone peptides were probably lost during chromatography (as tryptic peptides are too short and hydrophilic). In addition, longer post-translationally modified histone peptides were probably lost during database searches. The authors can repeat the database search with acetylation and methylation to be set as variable modifications and with a higher number of missed cleavages.

Response:

Thank you for catching this. Yes, the IPs were conducted in a neutral buffer. We have removed this speculation from the revised MS, since we don't have evidence to support this statement.

We didn't search a database with acetylation and methylation because our samples were not enriched for these modifications.

6) In Line 395 the authors stated: "Our proteomic analysis of Arabidopsis ASF1 interacting proteins showed a robust interaction with AtNASP1, HIRA, and all the known members of the HIRA complex."

Line 399: "However, whereas yeast and metazoan ASF1 proteins shuttle histones to both HIRA and CAF-1, we did not detect any peptide from any known component of the Arabidopsis CAF-1 complex in our ASF1 spectrometry data."

At this point, I have to stress once again that proteomic data supporting the authors' claims have not been shown.

Response:

We thank the reviewer's suggestion. We have added all identified proteins and their corresponding statistics presented on the volcano plots into the source data. We got ~2000 protein hits in our IP-MS, from which only HIRA complex components, AtNASP, and TOUSLED were significantly enriched.

Reviewer #4 (Remarks to the Author):

The revised manuscript has been much improved. The authors have addressed most of my concerns. However, I could not find pictures of asf1a1b double, asf1a1b hira-1 triple, and their respective controls, which according to the authors had been included in Figure S2 as a response to my suggestion #3. It seems to me that the authors did not include them in the revision by mistake.

Response:

Thank you again for reviewing our manuscript, and thank you for catching this. We are very sorry about this mistake. The figure is now available as Figure S3 and is also pasted below.

Figure S3. Morphology of Col-0, *asf1a-1*, *asf1b-1*, *hira-1*, *asf1a1b*, *hira-1 asf1a-1*, *hira-1 asf1b-1*, and *hira-1 asf1a1b*.

Reviewer #3 (Remarks to the Author):

1) The authors enclosed supporting data for IP-MS results. Here, I do not understand why certain proteins (see tab. Fig. 2a SAINT - ATAO1|AO1, AT3G52150, ATP5, AT4G17560; and tab. Fig. 2b SAINT - UBQ7|RUB2, UBQ12, AT3G11630, RPL24A) are missing in the volcano plots. Even though the authors think those proteins are unrelated to the topic, they should keep the dots in the volcano plots and mention those proteins in the text. The positions of certain dots do not correspond to the values presented in the tab. Fig2a and b (e.g., the dot of UBN1 in Fig. 2a is below 5 while FC in the table is 167.5 (log₂-FC 7.4). The authors should double-check the data in volcano plots and correct them.

2) The authors added the description of MS analysis and database search. They also explained that fold changes were calculated from spectral counts and apologized for the mistake in previous versions of the manuscript. In previous round of the review, I asked the authors to add a formula for NSAF calculation and its description. However, it seems that this is not relevant anymore as the authors realized that they did not use NSAF values for data evaluation. The description of NSAF including the formula should be removed from the text not to confuse the readers.

3) I appreciate that the authors apologized for the mistake in the fractionation description and corrected the respective paragraph.

4) In the previous review round, I suggested researching the data with acetylation and methylation, which could increase the sequence coverage of histone proteins. The authors refused to do it because their samples were not enriched for these modifications. Here, I would like to mention that there is no need for acetylation or methylation enrichment in the case of histones as they are decorated with those PTMs with high frequency.

5) The authors provided PRIDE account details (PRIDE - Proteomics Identification Database (ebi.ac.uk), login: reviewer_pxd035578@ebi.ac.uk, passw: x0UXGDYD). I am really embarrassed. **I found png files mainly of histone sequence coverage instead of MS raw data in the depository! - ??**

There is no need further to comment on it; I am just providing the link to Nature Communications guidance: nr-data-availability-statements-data-citations.pdf (nature.com):

"Data availability statements should provide a statement about the availability of data supporting the results reported in the article. By data we mean the minimal dataset that would be necessary to interpret, replicate and build upon the methods or findings reported in the article."

Screenshot of PRIDE database:

Project Files

Name	Type	Size (M)	Download
2017-10-10-140min-wb-Israel-Ausin-ASF1a-1.PNG	RAW	115944 bit	
2019-10-13-140min-200nl-yj-Zhenhui-Zhong-Steve-Jacobsen-Col-3-B.PNG	RAW	103104 bit	
2019-10-13-140min-200nl-yj-Zhenhui-Zhong-Steve-Jacobsen-Col-3-B.PNG	RAW	85271 bit	
2017-10-10-140min-wb-Israel-Ausin-ASF1a-3-B-2.PNG	RAW	87198 bit	
checksum.txt	OTHER	1846 bit	
2019-10-13-140min-200nl-yj-Zhenhui-Zhong-Steve-Jacobsen-Col-2-B.PNG	RAW	86078 bit	

Total 15 items < 1 > 20 /page v

An example of PNG file:

Peptide view

Project name:
Sample name:
Created date:

Filtered list file view | Filtered list file right and save | Unfiltered DTASelect list right click and save

Protein

accession	sequence coverage (%)	description
AT1G7560.1	6.62%	[Symbols Histone superfamily protein chr1:28390753-28391461 FORWARD LENGTH=136
AT4G40030.2	5.49%	[Symbols Histone superfamily protein chr4:18555845-18556827 REVERSE LENGTH=164
AT4G40040.1	6.62%	[Symbols Histone superfamily protein chr4:18557339-18558236 REVERSE LENGTH=136
AT5G10860.1	6.62%	[Symbols Histone superfamily protein chr5:3472591-3473349 REVERSE LENGTH=136

Export options: CSV | Excel | HTML | PDF

Peptide

uniqueId	sequence	spec count	confidence (%)	scan	charge	evaluation	fileName	primary score	DeRGN	MHH(calculated)	MHH(measured)	m/z(calculated)	m/z(measured)	ppm	RetTime
	R YRPTVALR.E spectrum view	1	99.9	12739	2		2019-10-13-140min-200nl-yj-Zhenhui-Zhong-Steve-Jacobsen-ASF1B2-B	2.4679	0.455	1032.5946	1032.595	516.8010643164963	516.8011253515625	0.1	NA

Export options: CSV | Excel | HTML | PDF

Dear Editors. Below we have responded to the third round of comments from reviewer 3.

1) The authors enclosed supporting data for IP-MS results. Here, I do not understand why certain proteins (see tab. Fig. 2a SAINT - ATAO1|AO1, AT3G52150, ATP5, AT4G17560; and tab. Fig. 2b SAINT - UBQ7|RUB2, UBQ12, AT3G11630, RPL24A) are missing in the volcano plots. Even though the authors think those proteins are unrelated to the topic, they should keep the dots in the volcano plots and mention those proteins in the text. The positions of certain dots do not correspond to the values presented in the tab. Fig2a and b (e.g., the dot of UBN1 in Fig. 2a is below 5 while FC in the table is 167.5 (log2-FC 7.4). The authors should double-check the data in volcano plots and correct them.

Response:

We have used two methods to identify interacting proteins. Results for both methods are attached in source data. The main method we used is the LIMMA method. And all dots showing in volcano plots are from the LIMMA method. SAINT method is just used to cross check results from LIMMA method. All results from LIMMA method are shown in volcano plots. As shown in source data, Log2FC value for UBN1 is 4.34, -Log10FDR is 2.16, which is consistent with presented in Fig.2 a volcano plot. We put data (source data Fig. 2a, 2b) and volcano (main Fig. 2a, 2b) for ASF1A and ASF1B side by side as below.

logFC	logCPM	PValue	FDR	Symbol
6.134485	11.52869	7.44E-08	0.000151	SPT/ATSP7 ASF1A SGA2
7.000305	11.57919	2.40E-07	0.000244	HIRA
5.167299	10.89958	6.31E-07	0.000428	NASP
6.53167	11.18195	8.78E-06	0.004462	CABIN1
5.408003	10.09426	1.55E-05	0.006303	TOUSLED
4.336622	8.963378	2.02E-05	0.006863	UBN1

logFC	logCPM	PValue	FDR	Symbol
7.035239	11.96784	5.30E-09	1.15E-05	CABIN1
6.992396	12.7895	6.60E-08	5.25E-05	SGA1 ASF1B SG
5.633003	10.59358	8.68E-08	5.25E-05	UBQ12
5.677409	10.6498	9.72E-08	5.25E-05	UBQ7 RUB2
7.872681	12.95888	1.72E-07	7.42E-05	HIRA
6.083056	11.09641	5.11E-07	0.000184	TOUSLED
4.236346	9.101143	1.11E-06	0.000343	UBN2
5.54852	11.10568	1.29E-06	0.000347	UBN1

2) The authors added the description of MS analysis and database search. They also explained that fold changes were calculated from spectral counts and apologized for the mistake in previous versions of the manuscript. In previous round of the review, I asked the authors to add a formula for NSAF calculation and its description. However, it seems that this is not relevant anymore as the authors realized that they did not use NSAF values for data evaluation. The description of NSAF including the formula should be removed from the text not to confuse the readers.

Response:

Thank you for this suggestion. We have removed this part so that it won't confuse the readers.

3) I appreciate that the authors apologized for the mistake in the fractionation description and corrected the respective paragraph.

Response:

Thank you and again we apologize for this mistake.

4) In the previous review round, I suggested researching the data with acetylation and methylation, which could increase the sequence coverage of histone proteins. The authors refused to do it because their samples were not enriched for these modifications. Here, I would like to mention that there is no need for acetylation or methylation enrichment in the case of histones as they are decorated with those PTMs with high frequency.

Response:

Although we appreciate that this could be done with some effort, this is not something that our mass spec collaborator does routinely, so it would be a bit of an undertaking. In addition, we feel like the IP-MS is relatively small part of this paper, in which we performed very general IP-MS method to identify potential interacting proteins of ASF1A and ASF1B, and used it together with other data including genetics and ChIP-seq data to support our conclusions. We feel that acetylation or methylation data, while nice, is not critical to support the main purpose of this study.

5) The authors provided PRIDE account details (PRIDE - Proteomics Identification Database (ebi.ac.uk), login: reviewer_pxd035578@ebi.ac.uk, passw: x0UXGDYD). I am really embarrassed. I found png files mainly of histone sequence coverage instead of MS raw data in the depository! - ??

There is no need further to comment on it; I am just providing the link to Nature Communications guidance: nr-data-availability-statements-data-citations.pdf (nature.com):

“Data availability statements should provide a statement about the availability of data supporting the results reported in the article. By data we mean the minimal dataset that would be necessary to interpret, replicate and build upon the methods or findings reported in the article.”

Response:

All raw and processed data related to this paper have been uploaded to GEO or PRIDE. For the PRIDE data, we submitted “SEARCH” files and peptide view of some proteins. We have now submitted RAW files. All data is available at PRIDE (<https://www.ebi.ac.uk/pride/>) with the following login information: login: reviewer_pxd035578@ebi.ac.uk, passw: x0UXGDYD